# Changes in seam number and location induce holes within microtubules assembled from porcine brain tubulin and in *Xenopus* egg cytoplasmic extracts

Charlotte Guyomar[1], Clément Bousquet[1], Siou Ku[1], John M Heumann[2], Gabriel Guilloux[1], Natacha Gaillard[3], Claire Heichette[1], Laurence Duchesne[1], Michel O Steinmetz[3,4], Romain Gibeaux[1], Denis Chrétien[1]*

[1]Univ Rennes, CNRS, IGDR (Institut de Génétique et Développement de Rennes) - UMR 6290, F-35000, Rennes, France; [2]Department of Molecular, Cellular and Developmental Biology, University of Colorado Boulder, Boulder, United States; [3]Laboratory of Biomolecular Research, Division of Biology and Chemistry, Paul Scherrer Institute, Villigen, Switzerland; [4]University of Basel, Biozentrum, Basel, Switzerland

*For correspondence:
denis.chretien@univ-rennes1.fr

**Abstract** Microtubules are tubes of about 25 nm in diameter that are critically involved in a variety of cellular functions, including motility, compartmentalization, and division. They are considered as pseudo-helical polymers whose constituent αβ-tubulin heterodimers share lateral homotypic interactions, except at one unique region called the seam. Here, we used a segmented sub-tomogram averaging strategy to reassess this paradigm and analyze the organization of the αβ-tubulin heterodimers in microtubules assembled from purified porcine brain tubulin in the presence of GTP and GMPCPP, and in *Xenopus* egg cytoplasmic extracts. We find that in almost all conditions, microtubules incorporate variable protofilament and/or tubulin subunit helical-start numbers, as well as variable numbers of seams. Strikingly, the seam number and location vary along individual microtubules, generating holes of one to a few subunits in size within their lattices. Together, our results reveal that the formation of mixed and discontinuous microtubule lattices is an intrinsic property of tubulin that requires the formation of unique lateral interactions without longitudinal ones. They further suggest that microtubule assembly is tightly regulated in a cytoplasmic environment.

## Editor's evaluation

This study presents an important finding on the assembly of microtubules in vitro, revealing structural defects accumulation in the lattice especially at the seam, where tubulin mediates lateral interactions. These defects appear at a low frequency in *Xenopus* egg cytoplasmic extracts, suggesting that cellular components control the microtubule lattice. The evidence supporting the conclusions is compelling, with rigorous cryo-electron tomography and image analysis. The work will be of broad interest to cell biologists and biochemists working on microtubules.

## Introduction

The organization of the αβ-tubulin heterodimer within microtubules was originally inferred from the analysis of transmission electron microscopy images of negatively stained axonemal doublets (*Amos and Klug, 1974*). It was proposed that the tubulin subunits engage heterotypic lateral interactions

**Figure 1.** Organization of tubulin within microtubules. The αβ-tubulin heterodimers (α in cyan, β in yellow) alternate head-to-tail along protofilaments, 13 of which associate laterally to form the microtubule wall. (**A**) In the A-type lattice, the lateral contacts are made between heterotypic subunits (α-β, β-α) along the three-start helices. (**B**) In the B-type lattice, the lateral contacts are made between homotypic subunits (α-α, β-β), except at one unique region of the A-type called the seam. (**C**) Decoration of microtubules with kinesin-motor domains (orange) that bind to β-tubulin highlights the organization of the tubulin heterodimer within microtubules.

(α-β, β-α) in the complete 13 protofilaments A-microtubule, and homotypic ones (α-α, β-β) in the incomplete 10 protofilaments B-microtubule, giving rise to the concept of the A and B lattices (*Figure 1A and B*). However, using kinesin-motor domains that bind uniquely to β-tubulin (*Figure 1C*), it was shown later that in both the A and B microtubules of the doublet, tubulin heterodimers engage homotypic interactions of the B type (*Song and Mandelkow, 1995*), which is also the case in microtubules assembled in vitro from purified tubulin (*Crepeau et al., 1978*; *Song and Mandelkow, 1993*). Noticeably, for geometrical reasons (*McEwen and Edelstein, 1980*; *Wade and Chrétien, 1993*), microtubules organized with 13 protofilaments and three-start lateral helices should contain at least one 'seam' of the A-type (*Figure 1B*), which corresponds to our current view of microtubule lattice organization.

Multiple seams were first visualized by freeze-etching and rotary shadowing of microtubules assembled in vitro (*Kikkawa et al., 1994*). Using the same approach on cells treated with detergent to remove the membrane and decorate the microtubules with kinesin-motor domains, the authors provided the first evidence of a preferred B-lattice-type organization in cellulo and could visualize unique seams in cytoplasmic microtubules. But due to the limitation of the method and the small number of microtubules observed, they did not exclude the possibility of several seams in cellulo. Since then, several studies have revealed the presence of multiple seams in microtubules assembled in vitro, noticeably in the presence of the stabilizing drug Taxol (*Debs et al., 2020*; *des Georges et al., 2008*; *Howes et al., 2017*; *Sosa et al., 1997*). The predominance of B-type lateral contacts in cellulo was confirmed by cryo-electron tomography after detergent removal of the membrane and decoration with kinesin-motor domains, but with no detailed statistics (*McIntosh et al., 2009*). Therefore, it turns out that our knowledge on the organization of αβ-tubulin heterodimers within microtubules assembled in vitro in the absence of drug and in cellulo remains limited.

To gain a deeper understanding of microtubule lattice organization in vitro and in a cytoplasmic environment, we analyzed microtubules assembled from purified porcine brain tubulin in the presence of GTP, the slowly hydrolysable analogue GMPCPP, and in *Xenopus* egg cytoplasmic extracts. Microtubules were decorated with kinesin-motor domains and their binding pattern was analyzed using cryo-electron tomography followed by sub-tomogram averaging (STA). To this end, we specifically developed a segmented sub-tomogram averaging (SSTA) strategy, which allowed us to investigate the structural heterogeneity of individual microtubules. We find that in almost all conditions the seam number and location vary within individual microtubules, leaving holes of one to a few subunits in size within their wall. Microtubules assembled in a cytoplasmic environment are more regular, suggesting a tightly regulated process. Moreover, the formation of discontinuous mixed AB-lattices implies that

tubulin can engage unique lateral interactions without longitudinal ones at the growing tip, a process that accounts for the formation of holes within their wall during polymerization.

## Results

Microtubules were self-assembled in vitro from purified porcine brain tubulin in the presence of 1 mM GTP (*Figure 2—figure supplement 1A*) and kinesin-motor domains were added at the polymerization plateau right before vitrification of the specimen grids into liquid ethane (*Figure 2—figure supplement 1B*). Cryo-electron tomograms were acquired preferentially using a dual-axis strategy (*Guesdon et al., 2013*) so that all microtubules could be analyzed independently of their orientation with respect to the tilt axes (*Figure 2—figure supplement 1C*, *Figure 2—video 1*). The low magnifications used, between ×25,000 and ×29,000, allowed us to record long stretches of the microtubules, ~1–2 µm in length, to optimize the STA strategy along individual fibers.

### The number and location of seams vary within individual microtubules assembled from purified tubulin

We first processed entire microtubules present in the tomograms using an STA approach that retrieves small sub-volumes of ~50 nm³ in size at every kinesin-motor domain position (*Zabeo et al., 2018*; i.e., every ~8 nm; *Figure 2—figure supplement 2A*). The resulting 3D volumes clearly revealed the protofilament number and the organization of the kinesin-motor domains around the microtubule lattice (*Figure 2*, *Figure 2—video 2*). To model the underlying αβ-tubulin heterodimer lattice, we placed yellow spheres onto kinesin densities and cyan spheres in between. While the cyan and yellow spheres are not strictly placed on top of the α- and β-subunits, respectively (*Figure 1C*), this simplified modeling allowed us to describe their underlying organization within microtubules (*Figure 2B and C*). In agreement with previous studies performed on Taxol-stabilized microtubules (*Debs et al., 2020*; *des Georges et al., 2008*; *Howes et al., 2017*; *Kikkawa et al., 1994*; *Sosa et al., 1997*), we found that microtubules assembled in vitro from purified tubulin in the presence of GTP contained one to several A-lattice seams (*Figure 3*). However, we could frequently observe protofilaments with a much thinner appearance where the kinesin-motor domain periodicity was partly or completely lost (*Figure 3A and B*, *Figure 3—video 1*). We hypothesized that the appearance of such aberrant protofilaments resulted from the averaging of regions containing kinesin-motor domain densities with regions out of register. To explore this idea, we divided the model and motive list used to calculate the full volumes into shorter segments of ~125–180 nm in length and generated new volumes for these segments (*Figure 3B and C*, *Figure 2—figure supplement 2B*). Using this SSTA approach, we could identify regions where the seam number and/or location varied within individual microtubules. In the example shown in *Figure 3B*, the segment S1 contains five seams while S3 and S4 contain three seams. S2 still displays two aberrant protofilaments, indicating that the change in seam number occurred in this region. To confirm this hypothesis, we extracted the corresponding region in the raw tomogram that was further filtered by thresholding intensities in Fourier space to increase the signal-to-noise ratio (*Figure 4A*). Comparison between the kinesin-motor domain patterns in the sub-tomogram averages of segments S1 to S3 with the filtered S2 region confirmed that this latter constitutes a transition zone where the seam number changes from 5 to 3. Line plots along the registered protofilaments (*Figure 4B*) shows that the kinesin-motor domain periodicity becomes out of phase after the transition in the aberrant protofilaments, implying an offset of at least one monomer (or an odd number) before and after the transition, and hence the presence of holes within the microtubule lattice (*Figure 4C*). Analysis of 24 microtubules taken on 4 tomograms, representing 195 segments of ~160 nm length (i.e., 2664 lateral interactions), allowed us to characterize 119 lattice-type transitions with an average frequency of 3.69 µm⁻¹ (*Table 1*), but with a high heterogeneity. Some microtubules showed no or little lattice-type transitions (e.g., MT3 and MT4, *Figure 2—figure supplement 3*; MT16, MT21, and MT23, *Figure 2—figure supplement 4*), while others were heavily dislocated, with a lattice-type transition frequency as high as ~15 µm⁻¹ (e.g., MT13 and MT14, *Figure 2—figure supplement 4*).

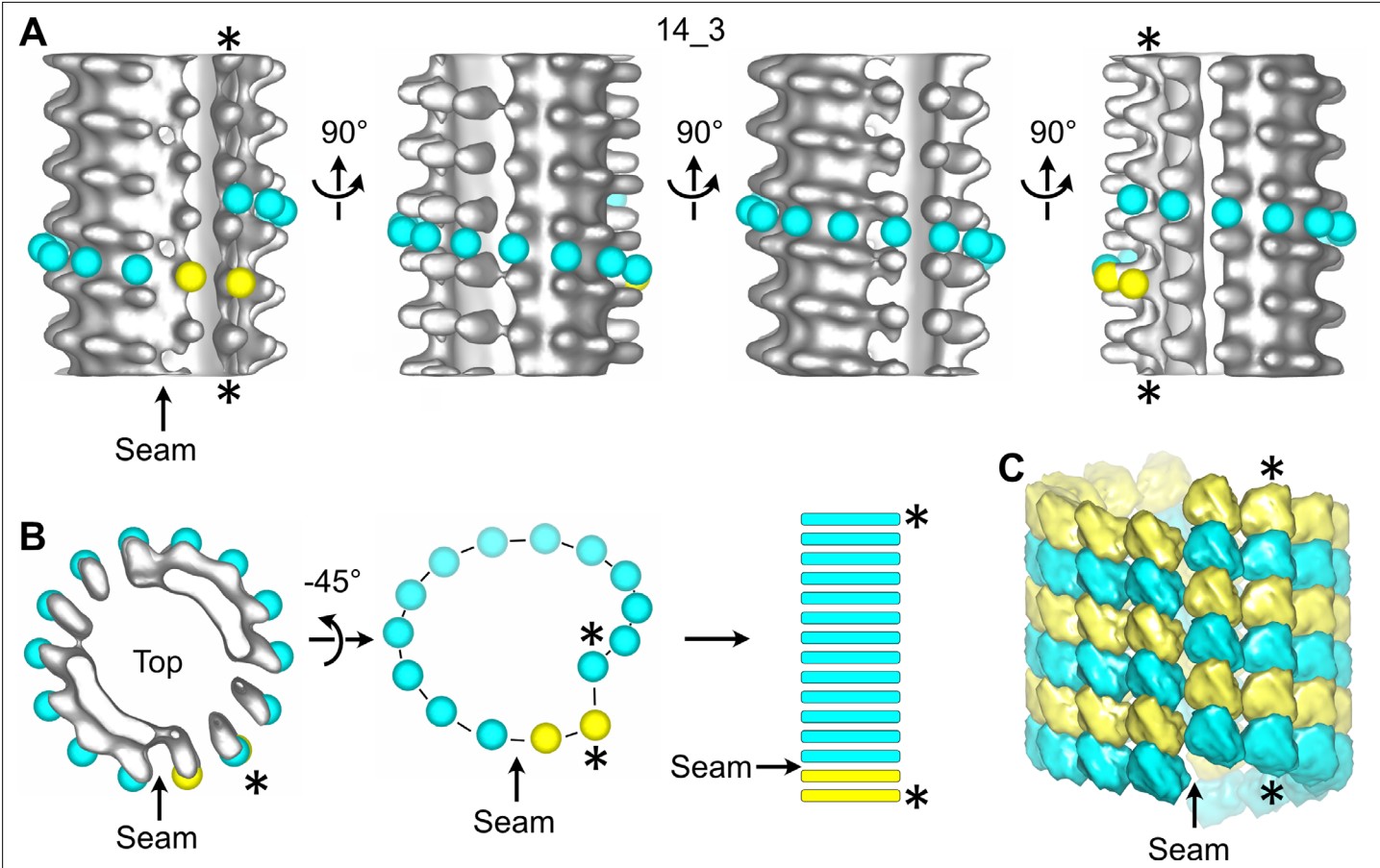

**Figure 2.** Sub-tomogram averaging of a 14_3 microtubule with a unique seam. (**A**) Sub-tomogram average of a 1390.4 nm long 14_3 microtubule assembled in vitro from purified tubulin and decorated with kinesin-motor domains (*Figure 2—figure supplement 3* MT3, *Figure 2—video 2*). The panel displays four views turned by 90° with respect to the longitudinal axis of the microtubule. Yellow spheres have been placed onto the kinesin-motor domain densities and cyan spheres in between them. They follow the left-handed, three-start helix of the microtubule lattice. The seam shows up as a change in color from yellow to cyan. (**B**) Symbolic representation of the microtubule lattice. The top view of the microtubule in (**A**) is turned by 45° around the X-axis, and the density is masked to reveal the organization of the tubulin subunits in one turn of the three-start helix. The helix is unrolled and represented as longitudinal bars that correspond to the organization of the αβ-subunits in microtubule segments. (**C**) 3D model of the underlying tubulin dimer lattice. The stars (*) indicate the same protofilament in (**A–C**).

The online version of this article includes the following video and figure supplement(s) for figure 2:

**Figure supplement 1.** Preparation of microtubules for cryo-electron tomography.

**Figure supplement 2.** Segmented sub-tomogram averaging (SSTA) of microtubules decorated with kinesin-motor domains.

**Figure supplement 3.** Lattice organization of microtubules assembled in vitro from purified porcine brain tubulin in the presence of GTP.

**Figure supplement 4.** Lattice organization of microtubules assembled in vitro from purified porcine brain tubulin in the presence of GTP.

**Figure 2—video 1.** Dual-axis cryo-electron tomography of microtubules.

https://elifesciences.org/articles/83021/figures#fig2video1

**Figure 2—video 2.** Sub-tomogram average of a 14_3 mono-seam microtubule.

https://elifesciences.org/articles/83021/figures#fig2video2

## Lattice-type transitions involve the formation of holes within microtubules

Direct visualization of holes within microtubules self-assembled at a tubulin concentration of 40 μM in the presence of GTP was hampered by the high background generated by free tubulin in solution. In addition, the low magnification used to analyze long stretches of the microtubules was at the detriment of resolution. To improve the quality of the raw cryo-electron tomograms, we used GMPCPP to assemble microtubules at a lower tubulin concentration (10 μM) and acquired single-axis

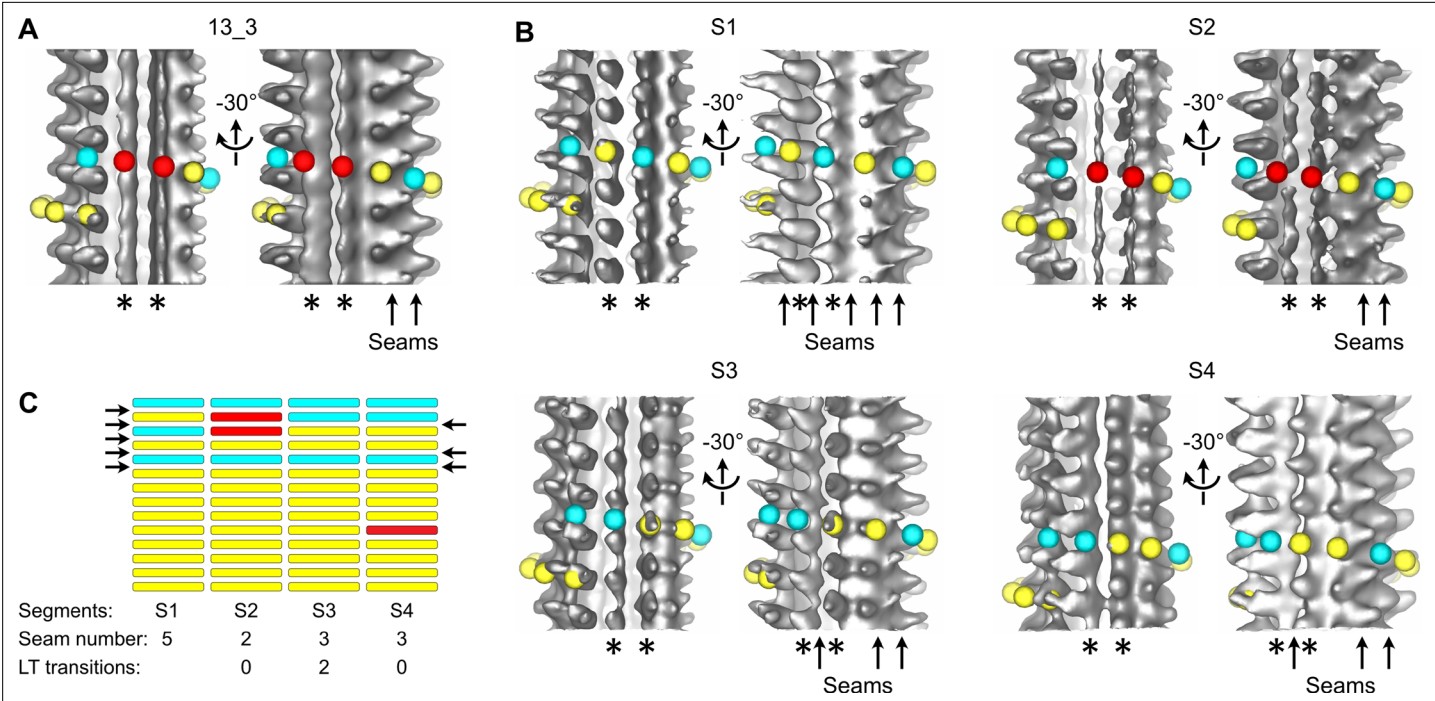

**Figure 3.** Transition in seam number within a 13_3 microtubule. (**A**) Average of a 1327.2-nm-long 13_3 microtubule displaying two aberrant protofilaments (*) and two adjacent seams (arrows in the –30° view). Red spheres have been placed on top of the aberrant protofilaments. (**B**) Segmented sub-tomogram averaging of the microtubule in (**A**). The microtubule has been divided into four segments of 331.8 nm in length, and sub-tomogram averages have been calculated for each segment (S1 to S4). The two aberrant protofilaments in (**A**) are well resolved in S1, S3, and S4, while they still display an aberrant shape in S2. The lattice organization of these protofilaments must be offset by at least one tubulin subunit between S1 and S3. Hence, S2 constitutes a transition zone where kinesin-motor domain densities and absence of densities have been averaged. (**C**) Flat representation of the lattice organization within segments S1 to S4. S1 contains five seams while S3 and S4 contain three seams (arrows). Two lattice-type (LT) transitions occur between S1 and S3, and S4 contains an aberrant protofilament (*Figure 3—video 1*). A finer segmentation of the microtubule at 165.9 nm revealed an additional lattice-type transition in this region (*Figure 2—figure supplement 3* MT5, between segments S5 and S7).

The online version of this article includes the following video for figure 3:

**Figure 3—video 1.** Segmented sub-tomogram averaging (SSTA) of a 13_3 multi-seam microtubule.

https://elifesciences.org/articles/83021/figures#fig3video1

tilt series at a magnification of ×50,000. SSTA was performed on kinesin-motor domains decorated GMPCPP-microtubules suitably oriented with respect to the tilt axis in order to localize transition regions and to visualize corresponding holes in their lattice. The microtubule shown in *Figure 5A* transitioned from 1 to 3 seams as demonstrated by SSTA. Visualization of the microtubule in the raw tomogram reveals a transition from a B- to an A-lattice organization in the three protofilaments located at its lower surface (*Figure 5B*), as assessed by the diffraction patterns of the corresponding regions, and after filtration of the equatorial and 8 nm⁻¹ layer lines. Enlargement of the central region (*Figure 5C*) shows a hole of one subunit's size in the middle protofilament (2) that accounts for the change in lattice organization at this location. In addition, the first protofilament (1) displays a gap of one dimer's size, although we cannot exclude that this results from an absence of kinesin-motor domain. Analysis of 31 GMPCPP-microtubules taken on six tomograms, representing 338 segments of ~150 nm in length (i.e., 3236 lateral interactions), and using the same strategy as in the presence of GTP (*Figure 5—figure supplements 1 and 2*) revealed a transition frequency of 1.25 μm⁻¹ (*Table 1*), that is, approximately threefold lower than microtubules assembled in the presence of GTP. However, since we used different tubulin concentrations, that is, 10 μM and 40 μM in the presence of GMPCPP and GTP, respectively, we cannot exclude a concentration-dependent effect on the lattice-type transition frequency.

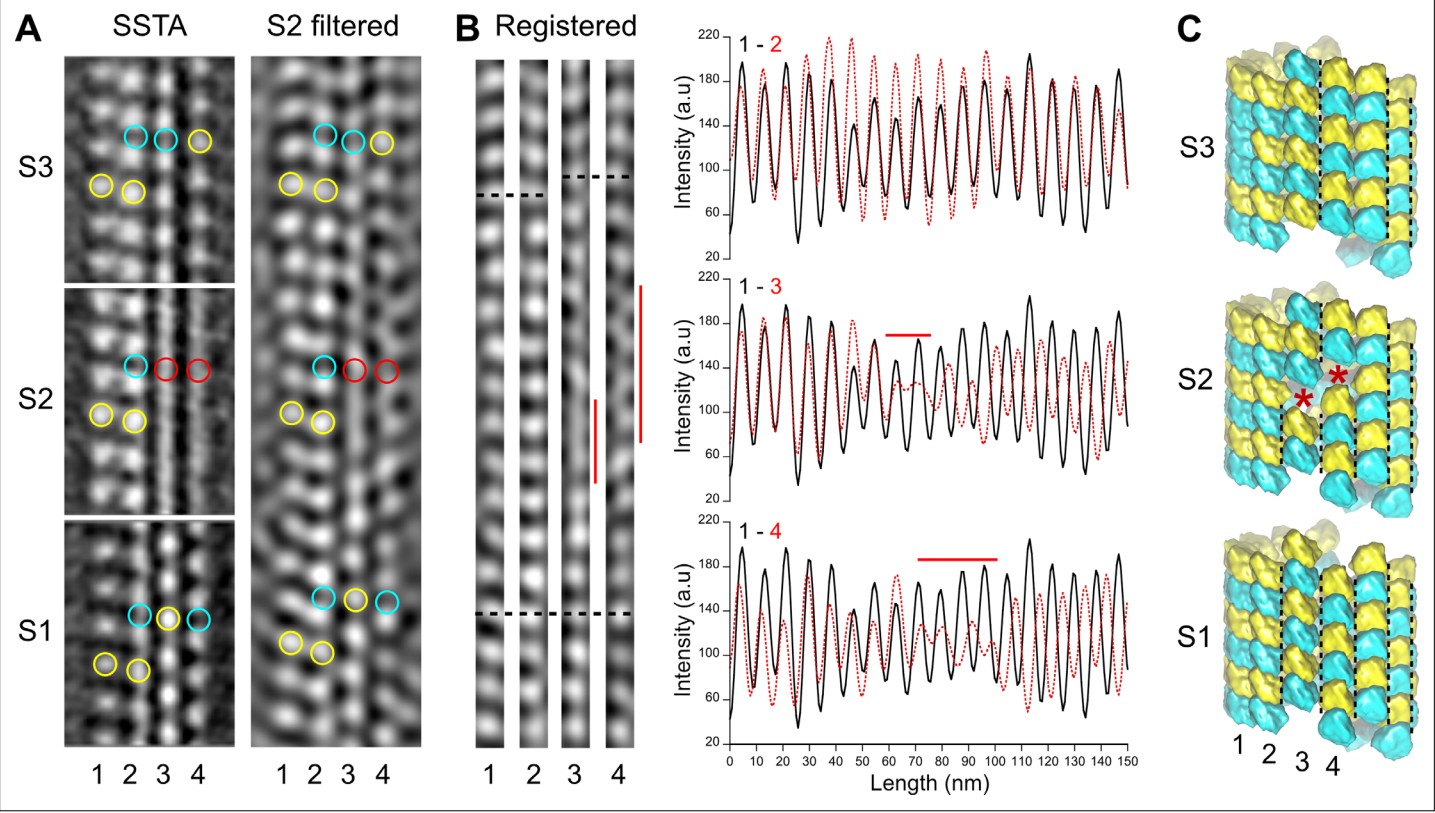

**Figure 4.** Comparison between segmented sub-tomogram averaging (SSTA) and Fourier filtered images of transition regions. (**A**) SSTA: slices through the sub-tomogram averages of segments S1 to S3 in *Figure 3B* (left). The contrast has been inverted with respect to the original tomogram to represent protein densities as white. Yellow open circles have been placed on kinesin-motor domain densities, cyan open circles in between them, and red open circles on aberrant densities in S2. S2 filtered: slice through the filtered tomogram of the S2 region (right). The change in lattice seam number from S1 to S3 is clearly visualized in the S2 region. (**B**) Protofilaments 1–4 in (**A**) have been extracted from the filtered image and put into register (Registered, left). They remain in phase (bottom dotted line) until the densities in protofilaments 3 and 4 becomes fuzzy (vertical red lines). After these transition zones, the kinesin-motor domain periodicity in protofilaments 3 and 4 becomes out of phase with respect to that in protofilaments 1 and 2 (upper dotted lines). These changes in kinesin-motor domain periodicity are confirmed in the line plots of the protofilaments (right). While the kinesin-motor domain periodicity in protofilaments 1 and 2 remain perfectly in phase (upper graph), it becomes out of phase for protofilaments 3 and 4 after the transition zones (middle and bottom graphs). (**C**) Schematic representation of the αβ-tubulin heterodimer organization in segments S1 to S3. The transition from five seams in S1 to three seams in S3 requires an offset of at least one monomer (red stars) in the protofilaments 3 and 4 of S2, although larger holes of an odd number of subunits could be present. Black dotted lines highlight the seams in each segment.

## Methodological artifacts limit the visualization of holes within microtubules in raw cryo-electron tomograms

During this study, we found strong limitations to the observation of holes within microtubules in raw tomograms. First, the transition regions had to be located at the top or bottom surface of the microtubule with respect to the electron beam since edges were severely smoothed due to the lack of data at high angle that elongate densities in this direction (*Figure 6A and B*). This artifact is inherent to electron tomography, limiting the search of holes within microtubules in raw tomograms. A second severe artifact commonly encountered, especially in thin ice layers, was denaturation of kinesin heads at the air–water interface (*Figure 6C and D*). This artifact shows up as a diminution of the kinesin-motor domain density, whose periodical arrangement can only be recovered after SSTA (*Figure 6D*). This analysis clearly showed that SSTA remains compulsory to localize changes in lattice-type organization within individual microtubules, and thus visualize the corresponding holes in regions suitably oriented with respect to the tilt axis and not in interaction with the air–water interfaces.

## Lattice-type transitions occur in a cytoplasmic environment

Next, we wondered whether the formation of holes was an intrinsic property of tubulin polymerization and whether such microtubule lattice defects were also present in a cellular context. Decorating

**Table 1.** Characterization of microtubule lattice structure by cryo-electron tomography and segmented sub-tomogram averaging.

| Assembly conditions | GTP | GMPCPP | Xenopus DMSO | Xenopus RanQ69L |
|---|---|---|---|---|
| Tomograms | 4 | 6 | 5 | 1 |
| Samples | 2 | 2 | 1 | 1 |
| Microtubules | 24 | 31 | 64 | 15 |
| Total length (µm) | 31.7 | 35.1 | 67.4 | 19.9 |
| Segments | 195 | 238 | 419 | 86 |
| Lateral interactions | 2663 | 3236 | 5446 | 1118 |
| A-type | 460 | 261 | 415 | 84 |
| B-type | 2091 | 2937 | 5025 | 1018 |
| ND | 112 | 38 | 6 | 16 |
| Lattice-type transitions | 119 | 37 | 6 | 2 |
| Frequency (µm$^{-1}$) | 3.69 ± 4.20 | 1.25 ± 1.41 | 0.09 ± 0.30 | 0.10 ± 0.29 |

microtubules with kinesin-motor domains in cells remains challenging since it involves removing of the cell membrane with detergents, adding kinesin-motor domains, and obtaining specimens thin enough to be analyzed by electron microscopy (*Kikkawa et al., 1994*; *McIntosh et al., 2009*). To overcome these difficulties and allow the analysis of a large data set of cytoplasmic microtubules, we took advantage of the open cellular system constituted by metaphase-arrested *Xenopus* egg cytoplasmic extracts (*Gibeaux and Heald, 2019*). Microtubule assembly was triggered using either DMSO (*Sawin and Mitchison, 1994*) or a constitutively active form of Ran (RanQ69L, *Carazo-Salas et al., 1999*) to control for possible effects of DMSO. Cryo-fluorescence microscopy was initially used to optimize the density of microtubule asters onto electron-microscope grids (*Figure 7A*). For structural analysis, no fluorescent tubulin was added to cytoplasmic extracts and kinesin-motor domains were added right before vitrification (*Figure 2—figure supplement 1D*). Specimens were imaged using dual-axis cryo-electron tomography (*Figure 7B and C*, *Figure 7—video 1*) followed by SSTA. A total of 64 microtubules taken on five tomograms were analyzed in the *Xenopus*-DMSO data set (i.e., 419 segments from which we characterized 5446 lateral interactions), and 15 microtubules taken on one tomogram for the *Xenopus* Ran-data set (i.e., 86 segments from which we characterized 1118 lateral interactions) (*Table 1*).

The vast majority of the microtubule segments were organized according to 13 protofilaments, three-start helices in a B-lattice configuration with one single seam (*Figure 8A*, *Figure 8—figure supplements 1–4*, *Table 1*). Yet, lattice-type transitions were observed in six cases over the 64 microtubules analyzed in the DMSO sample (MT2, MT5, MT14, MT18, MT28, and MT56; *Figure 8—figure supplements 1–4*). Similarly, two lattice-type transitions were observed over 15 microtubules analyzed in the Ran sample (MT4 and MT10; *Figure 8—figure supplement 5*), showing that the presence of lattice-type transitions was independent of the method used to trigger microtubule aster formation. The transition lattice-type frequencies were ~0.1 µm$^{-1}$ (*Table 1*), that is, at least one order of magnitude less than with microtubules assembled from purified tubulin in the presence of GMPCPP and GTP. Strikingly, these transitions systematically involved a lateral offset of the seam by one protofilament (*Figure 8B*, *Figure 8—video 1*). In addition, variations in protofilament and helix-start numbers were also observed such as 12_2, 12_3, 13_4, and 14_3 microtubule-lattice regions, but uniquely in the *Xenopus*-DMSO sample (*Figure 9A–D*, *Figure 9—video 1*, *Figure 10A*, *Table 2*). Of note, the 12_2 and 13_4 microtubules showed a local dislocation in between two protofilaments, which is likely a response to the excessive protofilament skewing present in these microtubules (*Chrétien and Fuller, 2000*). The 12_2 microtubule contained two seams (*Figure 9A*), while the 13_4 microtubules had no seam (*Figure 9C*), and hence were fully helical at the tubulin dimer level (MT7 and MT8, *Figure 8—figure supplement 1*). Microtubules with protofilament numbers different than 13 were not observed in the *Xenopus*-Ran sample (*Figure 8—figure supplement 5*, *Figure 10B*, *Table 2*).

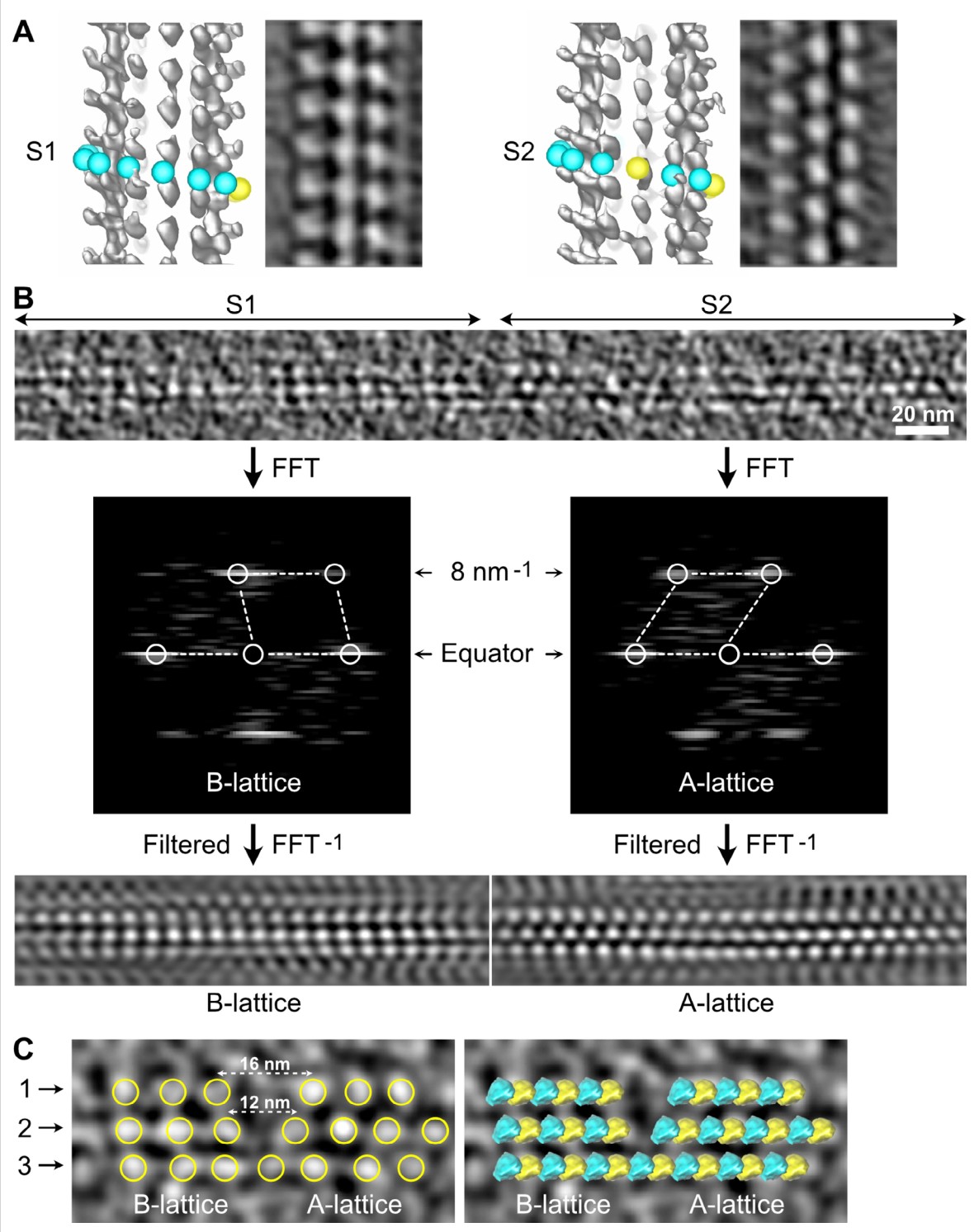

**Figure 5.** Direct visualization of holes within microtubules. (**A**) Sub-tomogram average segments before (S1) and after (S2) a lattice-type transition in a GMPCPP-microtubule. For each segment, the isosurface of the full volume (left) and a slice through the sub-tomogram average (right) are displayed. The contrast has been inverted to represent protein density as white. S1 and S2 contain 1 and 3 seams, respectively. (**B**) Z-projection of 20 slices at the surface of the microtubule that encompasses S1 and S2 (top) with their associated Fourier transforms (middle) and filtered versions of the corresponding regions after selection of the equatorial and 8 nm$^{-1}$ layer lines (bottom). The three protofilaments in S1 and S2 are organized according to a B- and an A-lattice, respectively. (**C**) Enlarged central region of the microtubule in (**B**). Yellow open circles have been placed on the kinesin densities (left), showing a gap of one subunit in protofilament 2, and possibly of a dimer in protofilament 1, although an absence of kinesin-motor domain at this location cannot

*Figure 5 continued on next page*

*Figure 5 continued*

be excluded. Tubulin heterodimers have been placed at the corresponding location (right) to highlight their change in organization at the transition region.

The online version of this article includes the following figure supplement(s) for figure 5:

**Figure supplement 1.** Lattice organization of microtubules assembled in vitro from purified porcine brain tubulin in the presence of GMPCPP.

**Figure supplement 2.** Lattice organization of microtubules assembled in vitro from purified porcine brain tubulin in the presence of GMPCPP.

Hence, we cannot exclude the possibility that DMSO induced the formation of these microtubules in *Xenopus* egg cytoplasmic extracts, and it remains to be determined whether they also occur in intact *Xenopus* eggs.

## Discussion

Here, we used a segmented sub-tomogram strategy to reveal changes in lattice types within individual microtubules assembled from purified tubulin or in a cytoplasmic context, and hence holes within their lattice. Ideally, cryo-electron tomography should reveal holes in the absence of averaging. Yet, we found severe limitations that are independent of the instrument used, but that are linked to the methodology. First, missing data at high angle, whether they are taken by single- or dual-axis cryo-electron tomography, blur densities on the edges of microtubules with respect to the tilt axis (*Guesdon et al., 2013*). Second, we found that the interaction of the microtubules with the air–water interface diminishes the kinesin-motor domain densities, likely as a consequence of denaturation (*D'Imprima et al., 2019*; *Klebl et al., 2022*). Third, the lattice-type transition frequency remains low with respect to the number of tubulin heterodimers within microtubules (*Figure 10C and D, Table 1*). It is 3.69 and 1.25 transitions every µm for microtubules assembled in the presence of GTP and GMPCPP, respectively, and 1 every ~10 µm for cytoplasmic extract microtubules. Considering that 1 µm of a 13 protofilament microtubule contains ~1625 dimers, this translates to one lattice-type transition every 16,250 dimers, which hinders the localization of holes in raw data. Conversely, the SSTA approach in combination with dual-axis cryo-electron tomography we used allows localization of lattice-type transitions along individual microtubules independently of their orientation with respect to the tilt axis, and at surfaces that interact with the air–water interface. While missing data at high angle are inherent to the method of electron tomography, means to limit denaturation of proteins at the air–water interface must be found. This is critical for cryo-electron tomography, but also for single-particle analysis methods where this artifact is a limiting factor to obtain high-resolution data (*Chen et al., 2019*; *Chen et al., 2022*; *D'Imprima et al., 2019*; *Klebl et al., 2022*; *Li et al., 2021*).

Changes in lattice types along individual microtubules could result from an imperfect annealing of shorter microtubules, a process known to occur in vitro (*Rothwell et al., 1986*). Yet, the lattice-type transition frequency observed with purified tubulin would necessitate annealing of very short segments, sometimes a few tens to hundreds of nm in length. The average lattice-type transition frequency observed in cytoplasmic extracts could be compatible with annealing of microtubules a few µm in length. However, the fact that these transitions involved systematically a lateral seam offset of only one protofilament suggests a firm regulatory mechanism. Hence, a more plausible explanation is that these lattice discontinuities are formed during microtubule assembly (*Figure 11*, *Figure 11—video 1*). At present, classical models of microtubule elongation hypothesize that tubulin engages either uniquely longitudinal interactions (*Figure 11A*, step 1), or both longitudinal and lateral interactions with the growing tip of microtubules (*Figure 11A*, step 2). A purely longitudinal elongation process (*McIntosh et al., 2018*) can hardly explain how microtubules can vary in terms of protofilament and/or helix start numbers as well as in lattice types, and thus how holes can arise during assembly. Conversely, to account for the presence of holes of one to a few subunits in size, it is sufficient to consider that tubulin can engage lateral interactions without longitudinal ones (*Figure 11A*, step 3). Gaps of an odd number of tubulin subunits will induce lattice-type transitions (*Figure 11A*, steps 4–5), while those of an even number will induce no changes (*Figure 11B*). Hence, since both types of events are likely to occur, we may underestimate the presence of holes within microtubules. In addition, a finer sampling of the microtubule lattice with shorter segments could also reveal a higher hole frequency. Formation of lateral contacts without longitudinal ones at the seam region

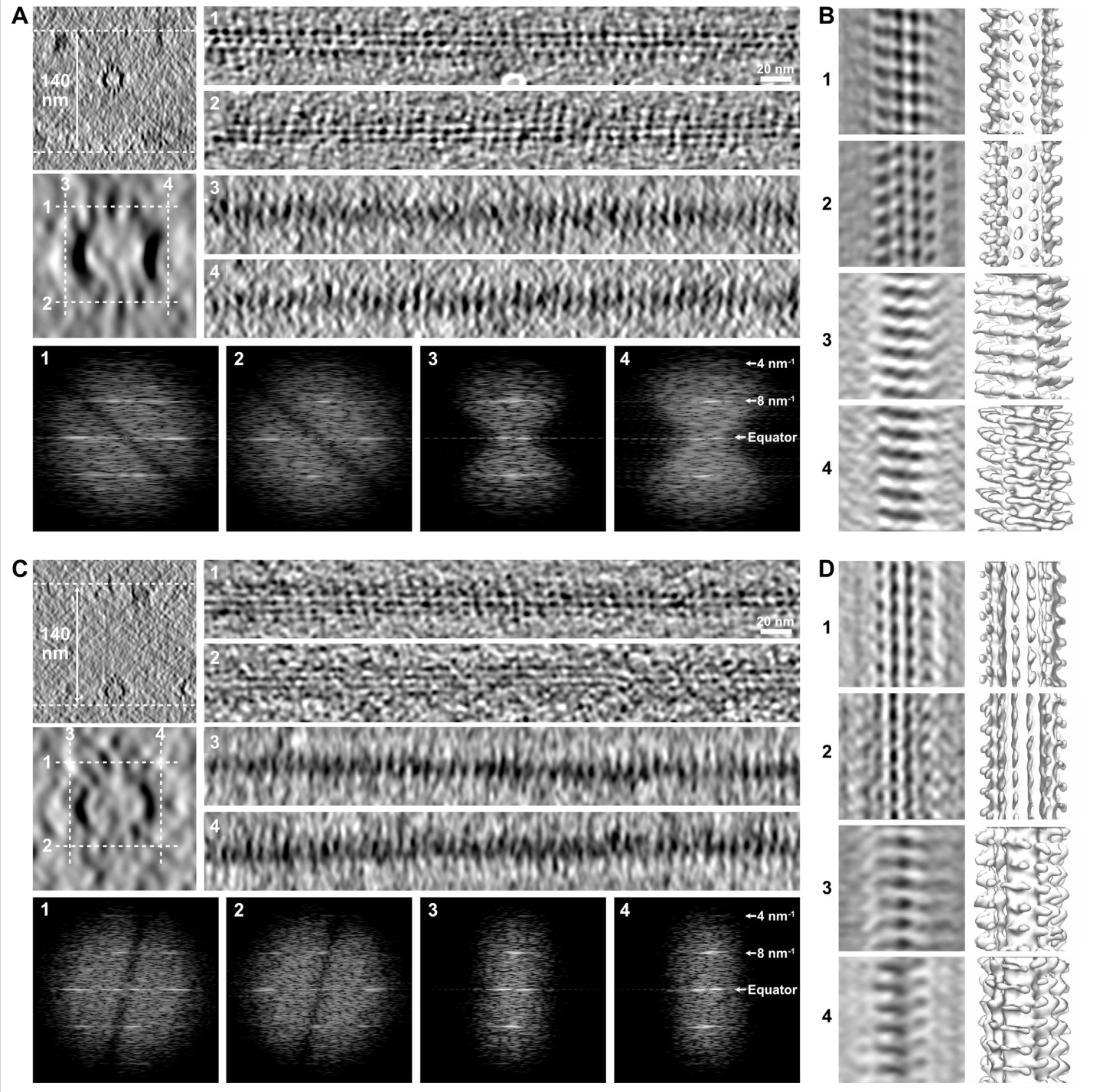

**Figure 6.** Limitations in the visualization of holes in raw tomograms. (**A**) Microtubule embedded in an ~140-nm-thick ice layer (top left). Longitudinal sections (averages of 20 slices, right) were performed at the top (1), bottom (2), left (3), and right (4) of the microtubule at positions indicated by white dotted lines in the enlarged view of the microtubule (middle left, average of 50 slices). Kinesin-motor domain densities can be individualized on the top (1) and bottom (2) sections, but not on the edges of the microtubules (3, 4) due to the elongation of densities in Z as a consequence of missing data at high angle. The Fourier transforms of the corresponding segments (bottom) show that the 8 nm$^{-1}$ periodicity of the kinesin-motor domains remains present in all views. (**B**) Sub-tomogram average of the microtubule in (**A**) over 18 kinesin-motor domain repeats. Sections (left) and isosurfaces (right) of the microtubule are displayed in correspondence to the longitudinal sections in (**A**). The kinesin-motor domain position is clearly observed on the top (1) and bottom (2) surfaces, and can be recovered on the microtubule edges after segmented sub-tomogram averaging (SSTA) (3, 4). (**C**) Microtubule in the same tomogram as in (**A**) interacting with the air–water interface (top left). Kinesin-motor domain densities can be well discerned on the longitudinal sections (right) of the top surface facing the solution (1), but are almost indiscernible on the bottom surface that interacts with the air–water interface

*Figure 6 continued on next page*

*Figure 6 continued*

(2) and on the edges (3, 4). Fourier transforms (bottom) of the corresponding segments show that the periodicity of the kinesin-motor domains is still present, even on the damaged surface (2). (**D**) Sub-tomogram average of the microtubule in (**C**) over 18 kinesin-motor domain repeats. SSTA allows recovery of the kinesin-motor domain densities in all surfaces, including the one that interacts with the air–water interface (2).

can also explain how the seam can vary in position by one protofilament (*Figure 11C*) since this only requires that a tubulin dimer engages homotypic lateral interactions at the seam region (*Figure 11C*, step 2). This event will also leave a gap of an odd number of subunits within the microtubule lattice (*Figure 11C*, steps 3–4).

Our current view of microtubules organized according to a perfect pseudo-helical B-lattice interrupted by a single A-lattice seam must be reconsidered. This is definitely the case for microtubules assembled from purified tubulin and has profound consequences for the interpretation of biochemical, biophysical, and structural results. For instance, 3D reconstruction studies will have to take into account the heterogeneity of the microtubule lattice to reach higher resolution (*Debs et al., 2020*). The lattice organization of cytoplasmic extract microtubules is more in agreement with the B-lattice, single-seam model. However, exceptions are also observed such as changes in protofilament and/or helix start numbers, as well as in the location of seams within individual microtubules. Therefore, our results suggest that the formation of heterogeneous microtubule lattices is an intrinsic property of tubulin polymerization, which is firmly regulated in cells. One key regulatory factor could be the γ-tubulin ring complex (γTuRC), which imposes the 13 protofilament organization to a nascent microtubule (*Böhler et al., 2021*). But how this structure is preserved during microtubule elongation remains unclear, especially if one considers a two-dimensional assembly process where the lattice can vary in terms of protofilament number, helix-start number, or lattice type during elongation. Proteins of the end-binding (EB) family are other good candidates that could play a key role in regulating microtubule structure during assembly in cells. They interact with the tip of growing microtubules and bind in between protofilaments that are organized according to a B-lattice *Maurer et al., 2012*; they thus may favor the formation of homotypic lateral interactions during assembly. In addition, EBs have been shown to induce the formation of 13 protofilaments, three-start helix microtubules (*Manka and Moores, 2018*; *Vitre et al., 2008*), which could also be forced to adopt a preferential B-lattice-type organization. Conversely, microtubule polymerases like XMAP215, which act at growing microtubule ends (*Brouhard et al., 2008*), may favor lattice heterogeneities (*Farmer et al., 2021*). It remains to be determined whether the concerted action of different microtubule growing-end binding proteins regulate microtubule structure and dynamics in cells (*Akhmanova and Steinmetz, 2008*).

## Ideas and speculation

Microtubules alternate stochastically between growing and shrinking states, an unusual behavior termed dynamic instability that was discovered some 38 years ago (*Mitchison and Kirschner, 1984*). Although it is exquisitely regulated in cells by a myriad of microtubule-associated proteins (*Cleary and Hancock, 2021*), it is also an intrinsic property of microtubules assembled from purified tubulin, demonstrating that it is intimately tied to tubulin assembly properties (*Brouhard, 2015*).

The αβ-tubulin heterodimer binds two molecules of GTP, one located between the α and β monomers at the non-exchangeable N-site, and one on the β-subunit at the longitudinal interface between heterodimers at the E-site that becomes hydrolyzed to GDP during assembly. GTP-hydrolysis destabilizes the microtubule lattice, likely weakening tubulin lateral interactions by a mechanism that remains unclear (*Zhang et al., 2015*). A slight delay between polymerization and GTP-hydrolysis would allow the formation of a protective GTP-cap at growing microtubule ends (*Pantaloni and Carlier, 1986*). The current model(s) speculate that stochastic loss of this GTP-cap induces depolymerization events known as catastrophes (*Mitchison and Kirschner, 1984*). However, the molecular mechanisms that lead to disappearance of the GTP-cap remain unknown. The origin of repolymerization events, termed rescues, is also unclear, but may involve tubulin molecules that have not hydrolyzed their GTP and that remain trapped inside the microtubule lattice (*Dimitrov et al., 2008*). It should be noted that the vast majority of theoretical models that have been designed so far to explain microtubule dynamic instability rely on a continuous lattice composed of B-type lattice contacts interrupted by a single seam of the A-type (*Bowne-Anderson et al., 2013*; *Bowne-Anderson et al., 2015*). Yet, exceptions to this rule have been documented over the years, essentially in microtubules assembled in vitro from

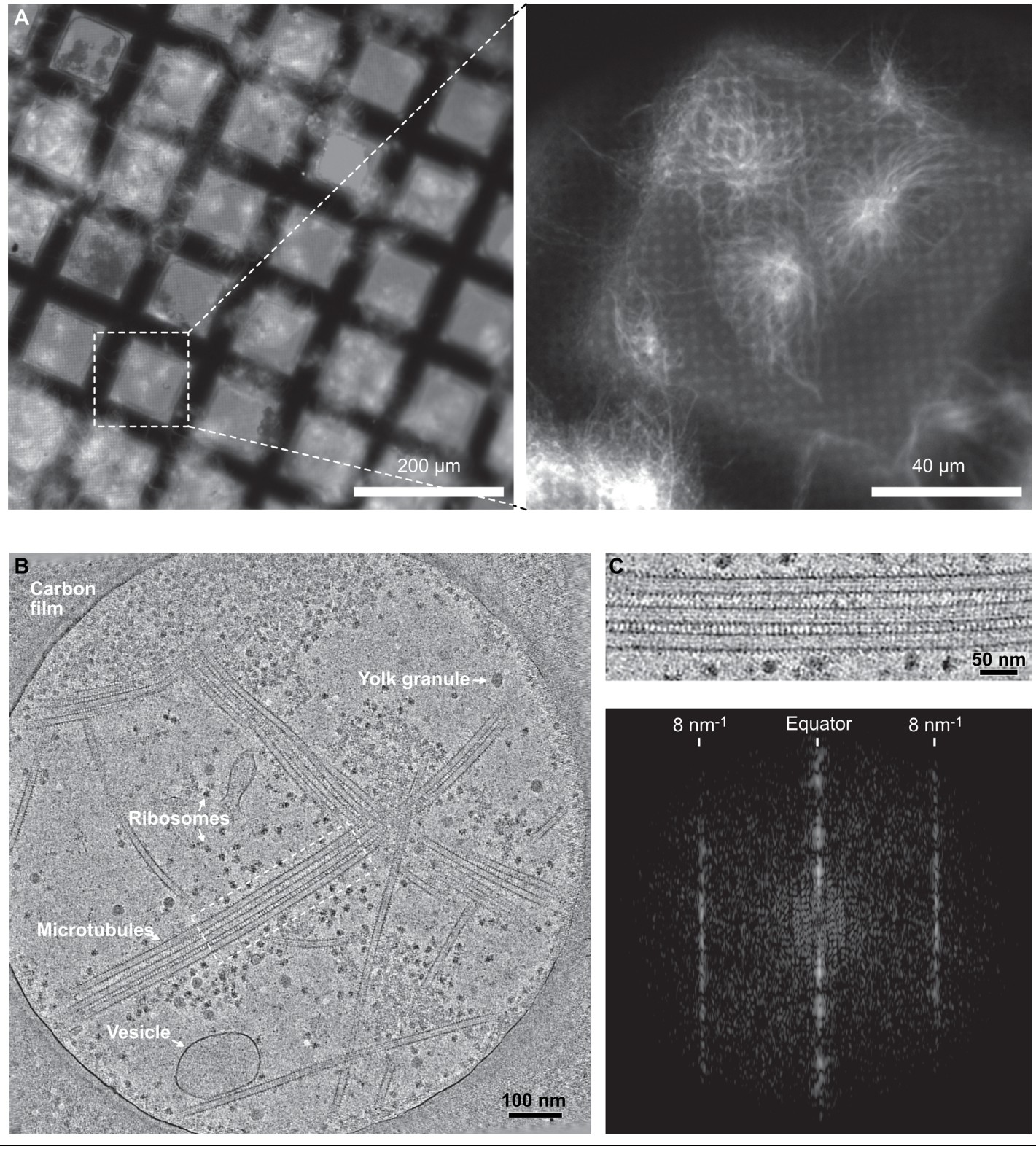

**Figure 7.** Cryo-electron tomography of microtubules decorated by kinesin-motor domains in *Xenopus* egg cytoplasmic extracts. (**A**) Cryo-fluorescence images of microtubules assembled in a cytoplasmic extract prepared from *Xenopus* eggs. Microtubules assembled in the presence of rhodamine-tubulin and plunge-frozen on an EM grid were imaged using fluorescence microscopy at liquid nitrogen temperature. Left: ×10 objective; right: ×50 objective. The white dashed square on the ×10 image indicates the field of view of the ×50 image. (**B**) Average of 30 slices in Z through a cryo-electron tomogram. The thin layer of cytoplasm spans a 2 µm diameter hole of the carbon film. The main visible features are ribosomes, vesicles, yolk granules,

*Figure 7 continued on next page*

*Figure 7 continued*

and microtubules decorated by kinesin-motor domains. (**C**) Top: enlargement of the dotted rectangular region in (**B**) (*Figure 7—video 1*). Bottom: Fourier transform of the top image showing strong layer lines at 8 nm⁻¹ corresponding to the kinesin-motor domain repeat along the microtubules.

The online version of this article includes the following video for figure 7:

**Figure 7—video 1.** Cytoplasmic extract microtubules decorated with kinesin-motor domains.

https://elifesciences.org/articles/83021/figures#fig7video1

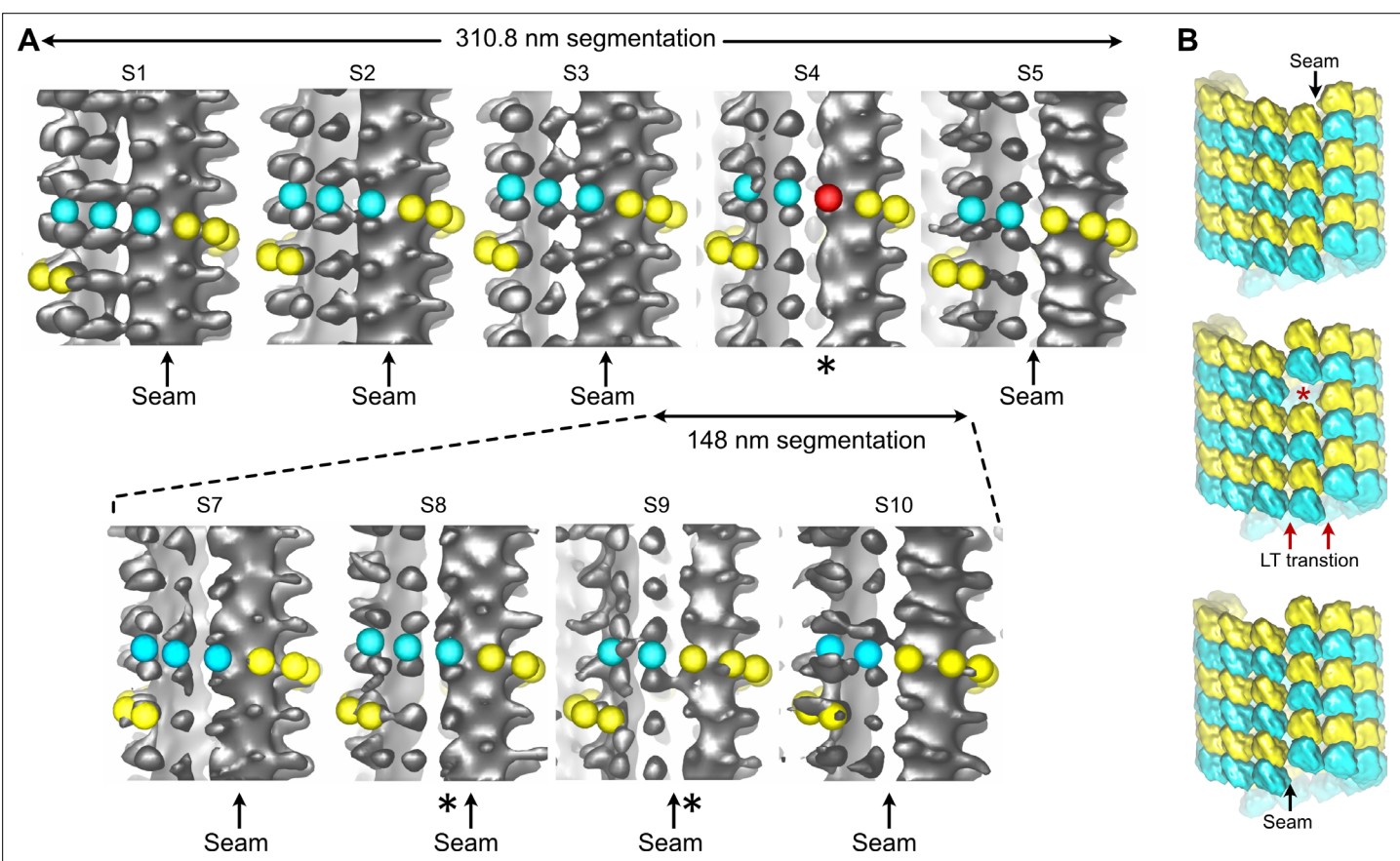

**Figure 8.** Segmented sub-tomogram averaging (SSTA) of microtubules decorated by kinesin-motor domains in *Xenopus* egg cytoplasmic extracts. (**A**) Sub-tomogram averages of five 400.5-nm-long segments of a 13_3 microtubule (top). S4 contains an aberrant protofilament (*), and the seam (arrow) moves laterally to the left by one protofilament from S3 to S5. The microtubule has been segmented into eleven 178-nm-long segments (bottom, *Figure 8—figure supplement 1*: MT2). Only S7 to S10 are shown, corresponding to a region that encompass S3 to S5 in the 310.8 nm segmentation (*Figure 8—video 1*). The lattice-type transition occurs from S8 to S9, and no aberrant protofilament is observed in this finer segmentation. (**B**) 3D models of the tubulin lattice before (top), during (middle), and after (bottom) the transition. The lateral offset in seam position requires a longitudinal offset of a minimum of one tubulin subunit to account for the lattice-type transition observed in (**A**).

The online version of this article includes the following video and figure supplement(s) for figure 8:

**Figure supplement 1.** Lattice organization of microtubules assembled in cytoplasmic Xenopus egg extracts in the presence of 5% DMSO.

**Figure supplement 2.** Lattice organization of microtubules assembled in cytoplasmic Xenopus egg extracts in the presence of 5% DMSO.

**Figure supplement 3.** Lattice organization of microtubules assembled in cytoplasmic Xenopus egg extracts in the presence of 5% DMSO.

**Figure supplement 4.** Lattice organization of microtubules assembled in cytoplasmic *Xenopus* egg extracts in the presence of 5% DMSO.

**Figure supplement 5.** Lattice organization of microtubules assembled in cytoplasmic *Xenopus* egg extracts in the presence of RanQ69L.

**Figure 8—video 1.** Segmented sub-tomogram averaging (SSTA) of a 13_3 microtubule assembled in a cytoplasmic extract.

https://elifesciences.org/articles/83021/figures#fig8video1

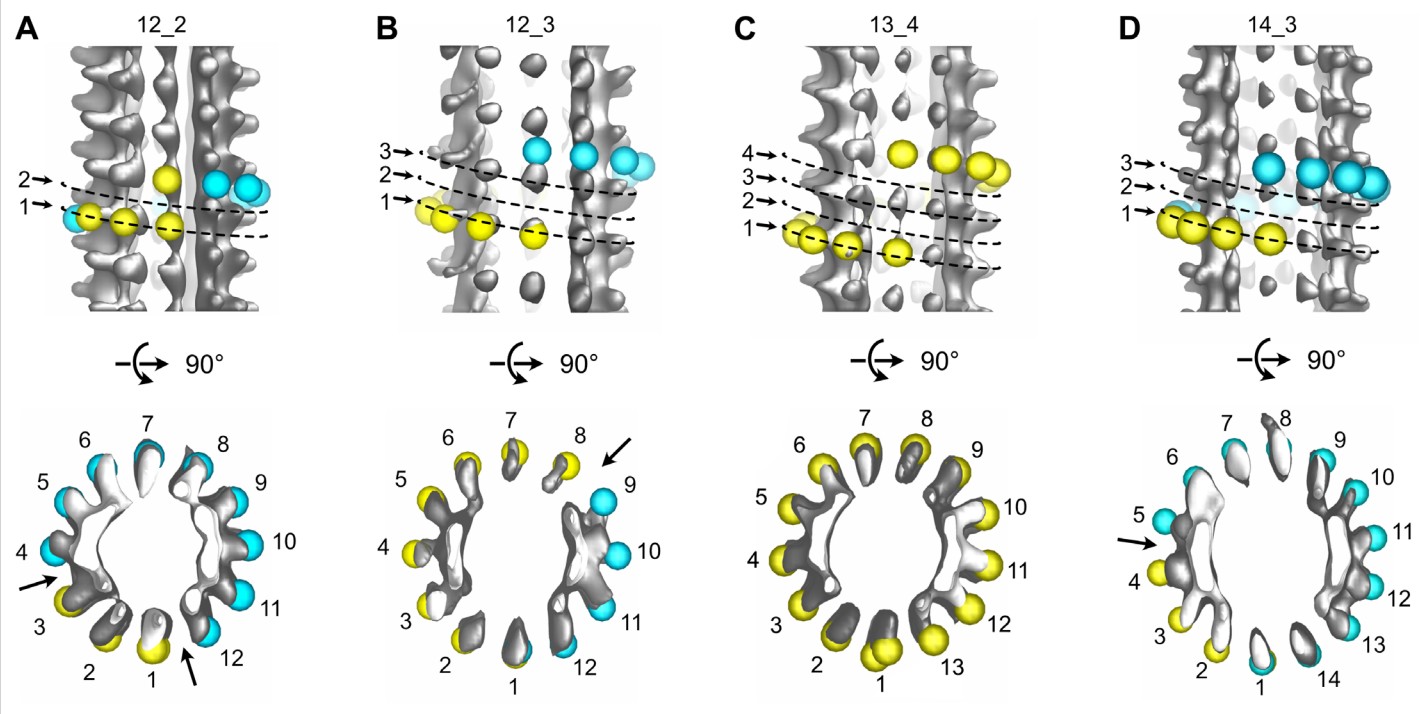

**Figure 9.** Variations in protofilament and helix-start numbers in microtubules assembled in *Xenopus* egg cytoplasmic extracts. (**A**) 12_2 microtubule with two seams (*Figure 8—figure supplement 1*: MT9). (**B**) 12_3 microtubule with a unique seam. This microtubule transitioned to a 13_3 configuration (*Figure 8—figure supplement 4*: MT62). (**C**) 13_4 microtubule with no seam. This microtubule transitioned to a 13_3 configuration (*Figure 8—figure supplement 1*: MT7). (**D**) 14_3 microtubule with one seam. This microtubule transitioned to a 13_3 configuration (*Figure 8—figure supplement 2* MT32).

The online version of this article includes the following video for figure 9:

**Figure 9—video 1.** Sub-tomogram averages of microtubules with different protofilament and/or helix-start numbers.

https://elifesciences.org/articles/83021/figures#fig9video1

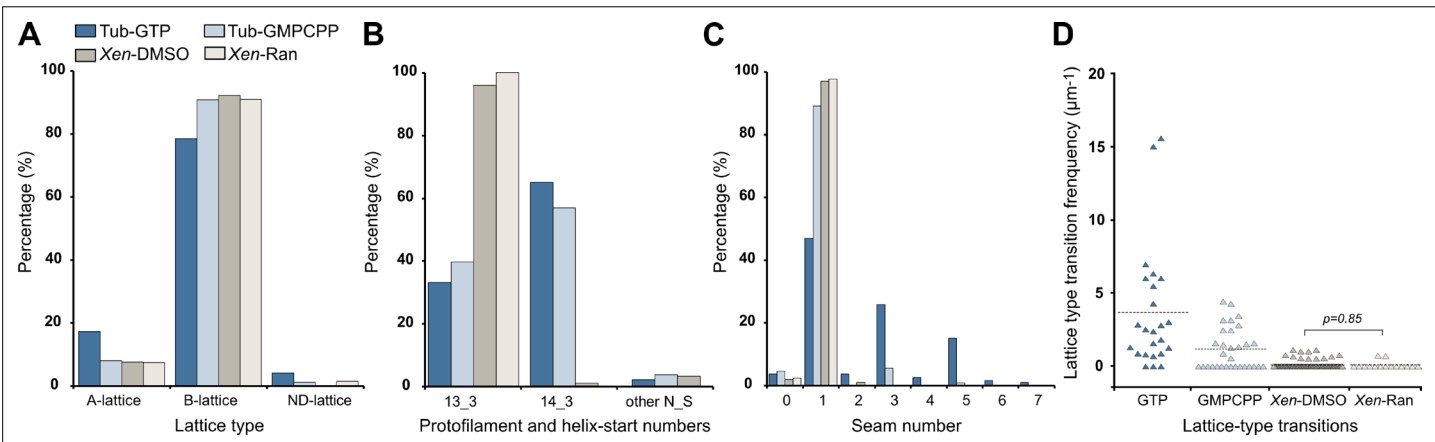

**Figure 10.** Characterization of microtubule lattices. (**A**) Percentage of lattice types. (**B**) Percentage of protofilament (N) and helix-start (S) numbers. (**C**) Percentage of seam number. (**D**) Lattice-type transition frequency. Microtubules assembled in *Xenopus* egg cytoplasmic extracts in the presence of 5% DMSO (*Xen*-DMSO) and RanQ69L (*Xen*-Ran) were compared using the Wilcoxon–Mann–Whitney rank-sum test. Tub-GTP: microtubules assembled at 40 µM tubulin concentration in the presence of 1 mM GTP. Tub-GMPCPP: microtubules assembled at 10 µM tubulin concentration in the presence of 0.1 mM GMPCPP.

**Table 2.** Protofilament (N) and helix-start number (S).

| N_S | 12_2 | 12_3 | 13_3 | 13_4 | 14_3 | 15_3 | 15_4 |
|---|---|---|---|---|---|---|---|
| GTP (%) | - | 1.00 | 32.99 | - | 64.97 | 1.04 | - |
| GMPCPP (%) | - | 1.88 | 39.65 | - | 56.76 | - | 1.71 |
| *Xen.* DMSO (%) | 1.11 | 0.21 | 95.97 | 1.78 | 0.93 | - | - |
| *Xen.* RanQ69L (%) | - | - | 100 | - | - | - | - |

Xen.: *Xenopus.*

purified tubulin. It is known that microtubules can accommodate different protofilament and helix-start numbers (*Chaaban and Brouhard, 2017*; *Chrétien and Wade, 1991*). These numbers can vary within individual microtubules (*Chrétien et al., 1992*; *Foster et al., 2022*), necessarily leaving holes inside their lattice (*Schaedel et al., 2019*; *Théry and Blanchoin, 2021*). Microtubules can also adopt configurations with a high protofilament skew that must be compensated by a relaxation step whose detailed mechanism remains to be described (*Chrétien and Fuller, 2000*). Microtubules with different numbers of seams have been described (*Debs et al., 2020*; *des Georges et al., 2008*; *Howes et al., 2017*; *Kikkawa et al., 1994*; *Sosa et al., 1997*), although it was not considered that both seam number and location could vary within individual microtubules. Therefore, these previous studies and this study indicate that the microtubule lattice is highly labile, with the ability to form different kinds of structural defects (*Hunyadi et al., 2005*; *Rai et al., 2021*).

The formation of lattice defects during microtubule polymerization must impose energetical penalties at the growing microtubule end, potentially destabilizing the protective GTP-cap if present, and hence be at the origin of catastrophes. Likewise, holes must let patches of unhydrolyzed GTP-tubulin molecules within microtubules, potentially at the origin of rescues. Hence, we propose that microtubule dynamic instability is not only driven by the nucleotide state of tubulin, but also by the intrinsic structural instability of the microtubule lattice. MAPs such as EBs and XMAP215 may exploit this structural instability to finely tune microtubule dynamics in cells.

## Materials and methods

### Key resources table

| Reagent type (species) or resource | Designation | Source or reference | Identifiers | Additional information |
|---|---|---|---|---|
| Strain, strain background (*Escherichia coli*) | Rosetta 2(DE3) | Merck | Cat# 71397-3 | Thermo-competent cells, used for kinesin-motor domain purification |
| Strain, strain background (*E. coli*) | One Shot BL21(DE3) | Thermo Fisher Scientific | Cat# C6000-03 | Used for the purification of GTPase-deficient mutant RanQ69L |
| Recombinant DNA reagent | KIF5B | Steinmetz laboratory (PSI Villigen, Switzerland) | | |
| Recombinant DNA reagent | RanQ69L | Heald laboratory (UC Berkley, USA) | | |
| Peptide, recombinant protein | Tubulin-Rhodamine | Cytoskeleton, Inc. | Cat# TL590M | Rhodamine labeled porcine brain tubulin |
| Chemical compound, drug | Chorulon 1500 | MSD Animal Health | GTIN: 08713184057587 | |
| Software, algorithm | ImageJ software | ImageJ (https://imagej.net/) | RRID:SCR_003070 | v1.53 |
| Software, algorithm | KaleidaGraph software | KaleidaGraph (https://www.synergy.com) | RRID:SCR_014980 | v5.01 |
| Software, algorithm | UCSF Chimera software | UCSF Chimera (https://www.cgl.ucsf.edu/chimera/) | RRID:SCR_004097 | v1.14 |
| Software, algorithm | IMOD software | IMOD (https://bio3d.colorado.edu/imod/) | RRID:SCR_003297 | v4.12.19 |
| Software, algorithm | PEET software | PEET (https://bio3d.colorado.edu/PEET/) | | v1.16.0 |

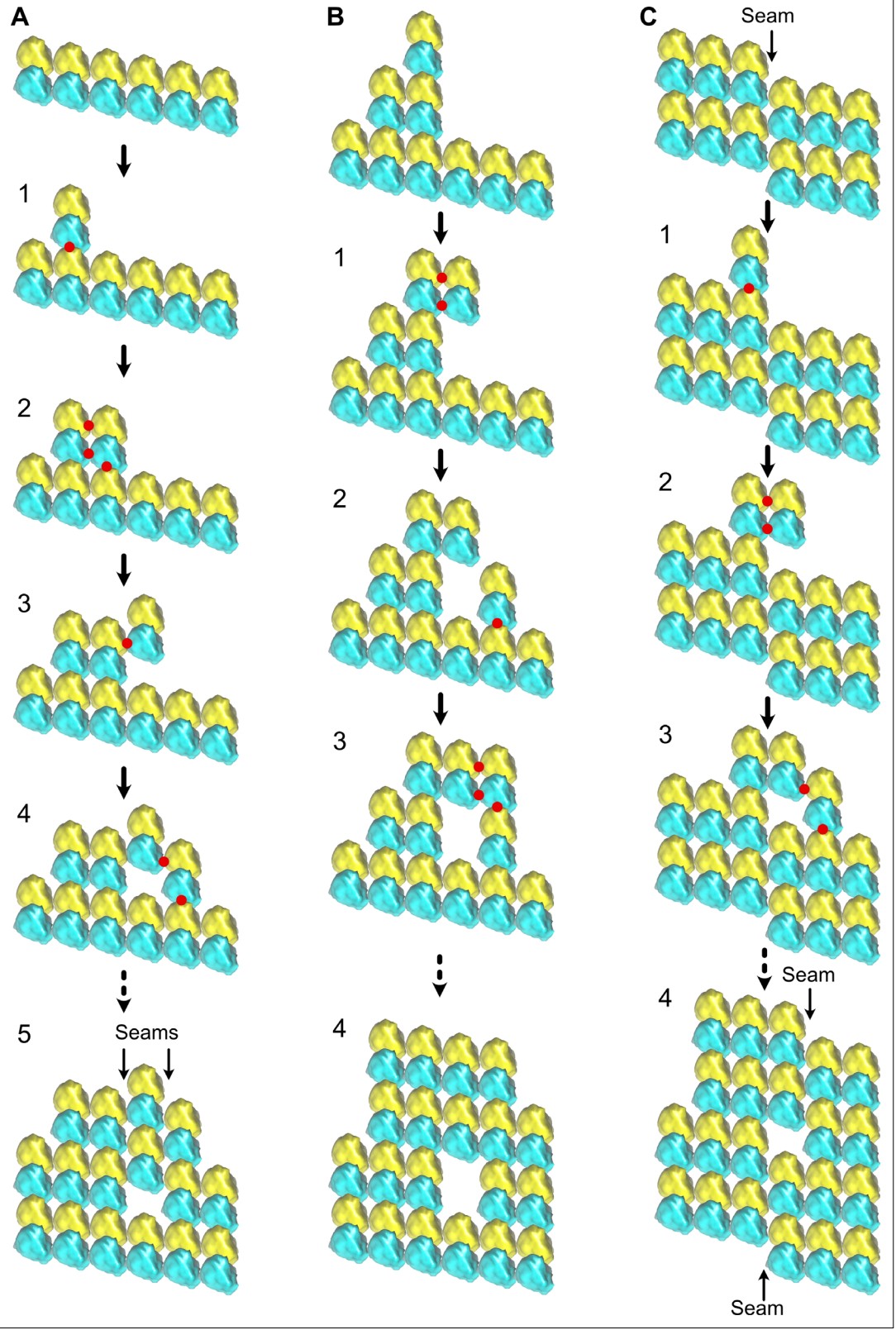

**Figure 11.** Formation of holes within microtubules during assembly. (**A**) Formation of multiple seams; red dots indicate new interactions. (1) Unique longitudinal interaction. (2) Combined lateral and longitudinal interactions. (3) Unique lateral interaction between one α-tubulin subunit of an incoming tubulin dimer and a β-tubulin subunit at the tip of the growing microtubule. (4–5) Incorporation of a hole within the microtubule lattice. Two A-lattice seams

*Figure 11 continued on next page*

*Figure 11 continued*

have been formed (arrows). (**B**) Incorporation of a tubulin dimer gap without change in lattice type organization. (1) Homotypic lateral interaction of an incoming tubulin dimer without longitudinal interaction. (2–5) Incorporation of a tubulin dimer gap inside the microtubule lattice. (**C**) Lateral offset of the seam by one protofilament during elongation. (1) Unique longitudinal interaction. (2) Homotypic interaction of an incoming dimer at the seam region without longitudinal contact. (3–4) Incorporation of a lattice-type transition inside the microtubule lattice. The seam has moved laterally by one protofilament (4), a situation systematically encountered in cytoplasmic extract microtubules.

The online version of this article includes the following video for figure 11:

**Figure 11—video 1.** Microtubule growth-induced mechanism of lattice heterogeneity.
https://elifesciences.org/articles/83021/figures#fig11video1

## Protein purification

Tubulin was isolated from porcine brain by two cycles of assembly disassembly (*Castoldi and Popov, 2003*), followed by a final cycle in the absence of free GTP (*Ashford and Hyman, 2006*). Tubulin was obtained in BRB80 (80 mM K-Pipes, 1 mM EGTA, 1 mM $MgCl_2$, pH 6.8 with KOH) and stored at –80°C before use.

The cDNA fragment encoding for the human Kif5B motor domain (residues 1–349) was cloned into the pET-based bacterial vector PSTCm1 (*Olieric et al., 2010*). The protein was expressed in Rosetta2 *Escherichia coli* cells. Cells were grown at 37°C in LB media supplemented with 50 µg/ml kanamycin and 30 µg/ml chloramphenicol to an $OD_{600}$ of 0.4–0.6. Temperature was reduced to 20°C, the protein production was induced 20 min later with 0.5 mM isopropy-1-thio-β-galactopyranoside (IPTG), and incubation was continued overnight under agitation. The cells were harvested by centrifugation for 15 min at 4000 × *g* and the cell pellets were resuspended in lysis buffer (50 mM HEPES, pH 8.0, supplemented with 10 mM imidazole, 10% glycerol, 0.1 mM ADP, 2 mM beta-mercaptoethanol, and one cOmplete EDTA free proteases inhibitor cocktail tablet per 50 ml buffer). The cells were lysed on ice per ultrasonication and lysate clearing was performed by centrifugation, 30 min at 24,000 × *g*. The resultant supernatant was filtered using a 0.45 µm filter and the protein was subsequently purified by IMAC on a 5 ml HP HisTrap column (GE Healthcare) according to the manufacturer's information. The eluted protein from this affinity step was concentrated and further purified by gel filtration on a HiLoad 16/600 Superdex 200 pg column (GE Healthcare) equilibrated in 20 mM Tris–HCl, pH 7.5, supplemented with 150 mM NaCl, 0.1 mM ADP, and 2 mM DTT. The homogeneity of the recombinant Kif5B motor domain was assessed by SDS-PAGE. Fractions were concentrated, aliquoted, flash-frozen into liquid nitrogen, and stored at –80°C.

## Animals

All animal experimentation in this study was performed according to our animal use protocol APAFiS #26858-2020072110205978 approved by the Animal Use Ethic Committee (#7, Rennes, France) and the French Ministry of Higher Education, Research and Innovation. Mature *Xenopus laevis* female frogs were obtained from the CRB Xénope (Rennes, France) and ovulated with no harm to the animals with at least a 6-month rest interval between ovulations.

## *Xenopus* egg cytoplasmic extracts

Cytostatic factor-arrested (CSF) egg extracts were prepared from freshly laid eggs of *X. laevis* as previously described (*Good and Heald, 2018*; *Murray, 1991*). Briefly, eggs arrested in metaphase of meiosis II were collected, dejellied, and fractionated by centrifugation. The cytoplasmic layer was isolated, supplemented with 10 mg/ml each of the protease inhibitors leupeptin, pepstatin, and chymostatin (LPC), 20 mM cytochalasin B, and a creatine phosphate and ATP energy regeneration mix. Vitrification of the samples for cryo-electron microscopy was performed the same day as the egg extract preparation.

## Cryo-fluorescence microscopy

To determine the optimal density of microtubule structures assembled from *Xenopus* egg cytoplasmic extracts cryo-fixed on electron microscopy grids suitable for cryo-electron tomography acquisitions,

we used cryo-fluorescence microscopy. Egg extracts were supplemented with 40 ng/µg rhodamine-tubulin (Cytoskeleton Inc, TL590M-B) before microtubule assembly was conducted by addition of 5% DMSO and incubation at 23°C, for 30–45 min. Reactions were then extemporaneously diluted 1:10, 1:50, or 1:100 in 1× BRB80 buffer for vitrification on an electron microscopy grid. Frozen grids were imaged within a Linkam CMS196M cryo-correlative microscopy stage mounted on an Olympus BX51 microscope equipped with a Lumencor SOLA SE U-nIR light source, UPLFLN10×/0.30 and LMPLFLN50×/0.50 objectives, and a Photometrics Prime-BSI sCMOS Back Illuminated camera. Images were acquired using the µManager acquisition software v1.4 (*Edelstein et al., 2014*).

## Cryo-electron tomography

Microtubules were assembled from purified porcine brain tubulin at 40 µM in BRB80, 1 mM GTP, or at 10 µM in BRB80, 0.1 mM GMPCPP, for about 1 hr at 35°C. Kif5B was diluted at a final concentration of 2.5 mg/ml in BRB80, 0.1 mM ATP, 1 mM GTP, and 60 nM mix-matrix capped gold nanoparticles (*Duchesne et al., 2008*; *Guesdon et al., 2016*) and prewarmed at 35°C. First, 3 µl of the microtubule sample was deposited at the surface of a glow-discharged holey carbon grid (Quantifoil R2/2, Cu200) in the temperature (35°C) and humidity-controlled atmosphere (~95 %) of an automatic plunge-freezer (EM GP, Leica). Then, 3 µl of the prewarmed kinesin-motor domain suspension was added to the grid onto the sample, incubated for 30 s, and blotted manually. An additional 3 µl of the prewarmed kinesin-motor domain suspension was added to the grid, blotted with the EM GP for 2 s using Whatman grade 1 filter paper and plunged into liquid ethane.

Microtubule aster assembly was induced in *Xenopus* egg cytoplasmic extracts by adding 5% DMSO or 15 µM of the GTPase-deficient Ran mutant RanQ69L purified as previously described (*Helmke and Heald, 2014*) and incubating at 23°C for 30–45 min. Kif5B was diluted at a final concentration of 2.5 mg/ml in BRB80, 0.1 mM ATP, 1 mM GTP, and 60 nM mix-matrix capped gold nanoparticles (*Duchesne et al., 2008*; *Guesdon et al., 2016*), and prewarmed at 23°C. A 3 µl volume of the Kif5B suspension was first deposited at the surface of a glow-discharged holey carbon grid (Quantifoil R2/2, Cu200) in the temperature (23°C) and humidity-controlled atmosphere (~95%) of the EM GP, on the side of the grid to be blotted. Immediately one volume of the extract sample was diluted 50× in the prewarmed Kif5B suspension, and 3 µl of this mix was deposited on the other side of the grid. The grid was blotted from the opposite side of the sample with the EM GP for 4 s using Whatman grade filter 4 and plunged into liquid ethane.

For dual-axis cryo-electron tomography, specimen grids were transferred to a rotating cryo-holder (model CT3500TR, Gatan) and observed using a 200 kV electron microscope equipped with a LaB$_6$ cathode (Tecnai G$^2$ T20 Sphera, FEI). Images of microtubules assembled from purified tubulin were recorded on a 4k × 4K CCD camera (USC4000, Gatan) in binning mode 2 and at a nominal magnification of ×29,000 (electron dose ~1.0 é/Å$^2$), providing a final pixel size of 0.79 nm. Images of microtubules assembled in *Xenopus* egg extracts were recorded on a 4K × 4k CMOS camera (XF416, TVIPS) in binning mode 2 and at a nominal magnification of ×25,000 (electron dose ~0.6 é/Å$^2$) or ×29,000 (electron dose ~1.0 é/Å$^2$), providing final pixel sizes of 0.87 nm and 0.74 nm, respectively. Pixel sizes were calibrated using TMV as a standard (*Guesdon et al., 2016*). Dual-axis cryo-electron tomography data were acquired as previously described (*Guesdon et al., 2013*). Briefly, a first tilt series of ~40 images was taken in an angular range of ~±60° starting from 0° and using a Saxton scheme. The specimen was turned by an ~90° in plane rotation at low magnification, and a second tilt series was taken on the same area using parameters identical to the first series. Tomograms were reconstructed using the Etomo graphical user interface of the IMOD program (*Kremer et al., 1996*; *Mastronarde, 1997*). Tilt series were typically filtered after alignment using a low-pass filter at 0.15 cycles/pixels and a sigma of 0.05. Tomograms were reconstructed in 3D using the SIRT-like filter of Etomo with 15 equivalent iterations. Dual-axis cryo-electron tomograms were converted to bytes before further processing.

For single-axis cryo-electron tomography, specimen grids were transferred to a dual-grid cryo-transfer holder model 205 (Simple Origin). Data were acquired on a 4K × 4k CMOS camera (XF416, TVIPS) in binning mode 1 and at a nominal magnification of ×50,000 (electron dose ~2.5 é/Å$^2$), providing a final pixel size of 0.21 nm. Typically, 40 images were taken in an angular range of ~±60° starting from 0° or using a symmetric electron dose scheme (*Hagen et al., 2017*). To localize holes within microtubules by SSTA, tomograms were subsequently binned by 4 to provide a final pixel size of 0.83 nm.

## Sub-tomogram averaging

Sub-tomogram averages were calculated using the procedure described online (available here). Briefly, a first model was created by following individual protofilaments in cross section using the slicer tool in IMOD. Usually, ~50 electronic slices were averaged to reinforce the contrast. A second model was next extrapolated from the first one to mark the microtubule center at the same point positions. Then, a third model was calculated from the previous ones with points spaced every ~8 nm, and a motive list containing Euler angles of each sub-volume with respect to the chosen reference was created. Sub-volumes of ~40 pixels$^3$ were extracted at each point position using the graphical user interface of the PEET program (*Nicastro et al., 2006*). Inner and outer cylindrical masks were used to isolate the microtubule wall densities. Registration of the microtubule sub-volumes was performed by cross-correlation, limiting rotational angular searches around the microtubule axis to about half the angular separation between protofilaments. Other angles were set to take into account variations of microtubule curvature in the X, Y, and Z directions. SSTA was performed using a new routine (splitIntoNsegments) implemented into the PEET program version 1.14.1. This routine splits the initial model and motive list into *N* segments of equal size and creates sub-directories for each segment. Sub-tomogram averages are calculated for each segment using the original sub-tomogram average parameters of the whole microtubule as a template.

## Image analysis and model building

Sub-tomogram averages were inspected using the isosurface panel of IMOD. Four scattered models were created. Model 1 was used to mark the kinesin-motor domain densities (yellow spheres), model 2 to mark the absence of densities (cyan spheres), model 3 aberrant densities (red spheres), and model 4 the microtubule center. Spheres from model 1–3 were placed along the *S*-start lateral helices. The fourth last model was enlarged to cross the kinesin-motor domain densities in order to place the other spheres at a same radius. The number of protofilaments and the different lateral contacts (A, B, and undefined lateral contacts) were retrieved from these models.

## Acknowledgements

Cryo-electron microscopy data were acquired on the Microscopy Rennes imaging center platform (Biosit, Rennes, France), member of the national infrastructure France-BioImaging (FBI) supported by the French National Research Agency (ANR-10-INBS-04). *Xenopus laevis* eggs were obtained from the Centre de Ressources Biologique *Xénopes*, Université de Rennes 1, Rennes, France. Porcine brains were kindly provided by Y Drillet, Cooperl Arc Altantique, Lamballe France. Tobacco Mosaic Virus was kindly provided by T Candresse, UMR 13332 Biologie du Fruit et Pathologie, INRAE and University of Bordeaux, Villenave d'Ornon, France. *Figure 11—video 1* was designed by A Kawska, Illuscienta, Paris, France. This work was supported by two French National Research Agency grants (ANR-16-CE11-0017-01 to DC and MOS, and ANR-18-CE13-0001-01 to DC), a Swiss National Science Foundation grant (310030_192566 to MOS), and a Human Frontier Science Program grant (CDA00019/2019C to RG).

## Additional information

### Funding

| Funder | Grant reference number | Author |
| --- | --- | --- |
| Agence Nationale de la Recherche | ANR-16-CE11-0017-01 | Denis Chrétien |
| Agence Nationale de la Recherche | ANR-18-CE13-0001-01 | Denis Chrétien |
| Human Frontier Science Program | CDA00019/1019-C | Romain Gibeaux |
| Swiss National Science Fondation | 310030_192566 | Michel O Steinmetz |

| Funder | Grant reference number | Author |
|---|---|---|

The funders had no role in study design, data collection and interpretation, or the decision to submit the work for publication.

## Author contributions

Charlotte Guyomar, Conceptualization, Software, Formal analysis, Supervision, Validation, Investigation, Visualization, Methodology, Writing - review and editing; Clément Bousquet, Software, Formal analysis, Investigation, Visualization, Methodology; Siou Ku, Formal analysis, Investigation, Visualization, Methodology; John M Heumann, Software, Methodology; Gabriel Guilloux, Investigation, Visualization; Natacha Gaillard, Claire Heichette, Laurence Duchesne, Resources, Methodology; Michel O Steinmetz, Romain Gibeaux, Conceptualization, Supervision, Funding acquisition, Writing - review and editing; Denis Chrétien, Conceptualization, Resources, Software, Formal analysis, Supervision, Funding acquisition, Validation, Investigation, Visualization, Methodology, Writing - original draft, Project administration, Writing - review and editing

## Author ORCIDs

Romain Gibeaux ⓘ http://orcid.org/0000-0001-5081-1985
Denis Chrétien ⓘ http://orcid.org/0000-0001-8261-4396

## Ethics

All animal experimentation in this study was performed according to our animal use protocol APAFiS #26858-2020072110205978 approved by the Animal Use Ethic Committee (#7, Rennes, France) and the French Ministry of Higher Education, Research and Innovation. Mature Xenopus laevis female frogs were obtained from the CRB Xénope (Rennes, France) and ovulated with no harm to the animals with at least a 6-month rest interval between ovulations.

## Decision letter and Author response

Decision letter https://doi.org/10.7554/eLife.83021.sa1
Author response https://doi.org/10.7554/eLife.83021.sa2

# Additional files

## Supplementary files

• Supplementary file 1. EMDB deposition IDs. Sub-tomogram averages and their corresponding models can be opened in Imod using the following command: 3dmod -V -E U emd_XXXX.map EMD-XXXX_helix.mod.

• Supplementary file 2. EMPIAR deposition IDs. To reconstruct the volumes, follow the instructions in the SSTA.txt document.

• MDAR checklist

• Source data 1. EMD_helix.zip: helix models associated with the sub-tomogram averages deposited on the EMDB (*Supplementary file 1*).

## Data availability

Sub-tomogram averages and extracts from cryo-electron tomograms presented in the figures have been deposited onto the EMDB and are listed in Supplementary File 1 with reference to the corresponding figures and videos. All the tilt series, tomograms, models and motive lists used to reconstruct the microtubule segments in PEET have been deposited onto the EMPIAR (Supplementary File 2).

The following datasets were generated:

| Author(s) | Year | Dataset title | Dataset URL | Database and Identifier |
|---|---|---|---|---|
| Guyomar C, Bousquet C, Ku S, Heumann J, Guilloux G, Gaillard N, Heichette C, Duchesne L, Steinmetz MO, Gibeaux R, Chrétien D | 2022 | Microtubule decorated with kinesin-motor domains, 14 protofilaments, 3-start helix, 1 seam | https://www.ebi.ac.uk/emdb/EMD-15735 | Electron Microscopy Data Bank, EMD-15735 |
| Guyomar C, Bousquet C, Ku S, Heumann J, Guilloux G, Gaillard N, Heichette C, Duchesne L, Steinmetz MO, Gibeaux R, Chrétien D | 2022 | Microtubule decorated with kinesin-motor domains; 13 protofilaments, 3-start helix, 3 seams, 2 abnormal protofilaments | http://ebi.ac.uk/emdb/EMD-15736 | Electron Microscopy Data Bank, EMD-15736 |
| Guyomar C, Bousquet C, Ku S, Heumann J, Guilloux G, Gaillard N, Heichette C, Duchesne L, Steinmetz MO, Gibeaux R, Chrétien D | 2022 | Microtubule decorated with kinesin-motor domains; 13 protofilaments, 3-start helix, 5 seams | https://www.ebi.ac.uk/emdb/EMD-15737 | Electron Microscopy Data Bank, EMD-15737 |
| Guyomar C, Bousquet C, Ku S, Heumann J, Guilloux G, Gaillard N, Heichette C, Duchesne L, Steinmetz MO, Gibeaux R, Chrétien D | 2022 | Microtubule decorated with kinesin-motor domains; 13 protofilaments, 3-start helix, 3 seams, 2 abnormal protofilaments | https://www.ebi.ac.uk/emdb/EMD-15738 | Electron Microscopy Data Bank, EMD-15738 |
| Guyomar C, Bousquet C, Ku S, Heumann J, Guilloux G, Gaillard N, Heichette C, Duchesne L, Steinmetz MO, Gibeaux R, Chrétien D | 2022 | Microtubule decorated with kinesin-motor domains; 13 protofilaments, 3-start helix, 3 seams | https://www.ebi.ac.uk/emdb/EMD-15739 | Electron Microscopy Data Bank, EMD-15739 |
| Guyomar C, Bousquet C, Ku S, Heumann J, Guilloux G, Gaillard N, Heichette C, Duchesne L, Steinmetz MO, Gibeaux R, Chrétien D | 2022 | Microtubule decorated with kinesin-motor domain, 13 protofilaments, 3-start helix, transition from 5 to 3 seams | https://www.ebi.ac.uk/emdb/EMD-15734 | Electron Microscopy Data Bank, EMD-15734 |
| Guyomar C, Bousquet C, Ku S, Heumann J, Guilloux G, Gaillard N, Heichette C, Duchesne L, Steinmetz MO, Gibeaux R, Chrétien D | 2022 | Microtubule decorated with kinesin-motor domains, 13 protofilaments, 3-start helix, 3 seams | https://www.ebi.ac.uk/emdb/EMD-15740 | Electron Microscopy Data Bank, EMD-15740 |

*Continued*

| Author(s) | Year | Dataset title | Dataset URL | Database and Identifier |
|---|---|---|---|---|
| Guyomar C, Bousquet C, Ku S, Heumann J, Guilloux G, Gaillard N, Heichette C, Duchesne L, Steinmetz MO, Gibeaux R, Chrétien D | 2022 | Microtubule decorated with kinesin-motor domains, 13 protofilaments, 3-start helix, 1 seam | https://www.ebi.ac.uk/emdb/EMD-15741 | Electron Microscopy Data Bank, EMD-15741 |
| Guyomar C, Bousquet C, Ku S, Heumann J, Guilloux G, Gaillard N, Heichette C, Duchesne L, Steinmetz MO, Gibeaux R, Chrétien D | 2022 | Microtubule decorated with kinesin-motor 3, 13 protofilaments, 3-start helix, transition from 3 to 1 seams | https://www.ebi.ac.uk/emdb/EMD-15742 | Electron Microscopy Data Bank, EMD-15742 |
| Guyomar C, Bousquet C, Ku S, Heumann J, Guilloux G, Gaillard N, Heichette C, Duchesne L, Steinmetz MO, Gibeaux R, Chrétien D | 2022 | Microtubule decorated with kinesin-motor domains, 13 protofilaments, 3-start helix, 1 seam, fully embedded in ice | https://www.ebi.ac.uk/emdb/EMD-15743 | Electron Microscopy Data Bank, EMD-15743 |
| Guyomar C, Bousquet C, Ku S, Heumann J, Guilloux G, Gaillard N, Heichette C, Duchesne L, Steinmetz MO, Gibeaux R, Chrétien D | 2022 | Microtubule decorated with kinesin-motor domains, fully embedded in ice | https://www.ebi.ac.uk/emdb/EMD-15750 | Electron Microscopy Data Bank, EMD-15750 |
| Guyomar C, Bousquet C, Ku S, Heumann J, Guilloux G, Gaillard N, Heichette C, Duchesne L, Steinmetz MO, Gibeaux R, Chrétien D | 2022 | Microtubule decorated with kinesin-motor domains, 13 protofilaments, 3-start helix, 1 seam, in interaction with the air-water interface | https://www.ebi.ac.uk/emdb/EMD-15751 | Electron Microscopy Data Bank, EMD-15751 |
| Guyomar C, Bousquet C, Ku S, Heumann J, Guilloux G, Gaillard N, Heichette C, Duchesne L, Steinmetz MO, Gibeaux R, Chrétien D | 2022 | Microtubule decorated with kinesin-motor domains, interacting with the air-water interface | https://www.ebi.ac.uk/emdb/EMD-15752 | Electron Microscopy Data Bank, EMD-15752 |
| Guyomar C, Bousquet C, Ku S, Heumann J, Guilloux G, Gaillard N, Heichette C, Duchesne L, Steinmetz MO, Gibeaux R, Chrétien D | 2022 | Microtubules assembled in *Xenopus* egg cytoplasmic extract and decorated with kinesin-motor domains | https://www.ebi.ac.uk/emdb/EMD-15732 | Electron Microscopy Data Bank, EMD-15732 |

*Continued on next page*

*Continued*

| Author(s) | Year | Dataset title | Dataset URL | Database and Identifier |
|---|---|---|---|---|
| Guyomar C, Bousquet C, Ku S, Heumann J, Guilloux G, Gaillard N, Heichette C, Duchesne L, Steinmetz MO, Gibeaux R, Chrétien D | 2022 | Microtubule decorated with kinesin-motor domains, 13 protofilaments, 3-start helix, 1 seam | https://www.ebi.ac.uk/emdb/EMD-15733 | Electron Microscopy Data Bank, EMD-15733 |
| Guyomar C, Bousquet C, Ku S, Heumann J, Guilloux G, Gaillard N, Heichette C, Duchesne L, Steinmetz MO, Gibeaux R, Chrétien D | 2022 | Microtubule decorated with kinesin-motor domains, 13 protofilaments, 3-start helix, 0 seam, 1 abnormal protofilament | https://www.ebi.ac.uk/emdb/EMD-15744 | Electron Microscopy Data Bank, EMD-15744 |
| Guyomar C, Bousquet C, Ku S, Heumann J, Guilloux G, Gaillard N, Heichette C, Duchesne L, Steinmetz MO, Gibeaux R, Chrétien D | 2022 | Microtubule decorated with kinesin-motor domains, 13 protofilaments, 3-start helix, 1 seam | https://www.ebi.ac.uk/emdb/EMD-15745 | Electron Microscopy Data Bank, EMD-15745 |
| Guyomar C, Bousquet C, Ku S, Heumann J, Guilloux G, Gaillard N, Heichette C, Duchesne L, Steinmetz MO, Gibeaux R, Chrétien D | 2022 | Microtubule decorated with kinesin-motor domains, 12 protofilaments, 2-start helix, 2 seams | https://www.ebi.ac.uk/emdb/EMD-15746 | Electron Microscopy Data Bank, EMD-15746 |
| Guyomar C, Bousquet C, Ku S, Heumann J, Guilloux G, Gaillard N, Heichette C, Duchesne L, Steinmetz MO, Gibeaux R, Chrétien D | 2022 | Microtubule decorated with kinesin-motor domains, 12 protofilaments, 3-start helix, 1 seam | https://www.ebi.ac.uk/emdb/EMD-15747 | Electron Microscopy Data Bank, EMD-15747 |
| Guyomar C, Bousquet C, Ku S, Heumann J, Guilloux G, Gaillard N, Heichette C, Duchesne L, Steinmetz MO, Gibeaux R, Chrétien D | 2022 | Microtubule decorated with kinesin-motor domains, 13 protofilaments, 4-start helix, 0 seam | https://www.ebi.ac.uk/emdb/EMD-15748 | Electron Microscopy Data Bank, EMD-15748 |
| Guyomar C, Bousquet C, Ku S, Heumann J, Guilloux G, Gaillard N, Heichette C, Duchesne L, Steinmetz MO, Gibeaux R, Chrétien D | 2022 | Microtubule decorated with kinesin-motor domains, 14 protofilaments, 3-start helix, 1 seam | https://www.ebi.ac.uk/emdb/EMD-15749 | Electron Microscopy Data Bank, EMD-15749 |

*Continued on next page*

*Continued*

| Author(s) | Year | Dataset title | Dataset URL | Database and Identifier |
|---|---|---|---|---|
| Guyomar C, Bousquet C, Ku S, Heumann J, Guilloux G, Gaillard N, Heichette C, Duchesne L, Steinmetz MO, Gibeaux R, Chrétien D | 2022 | Cryo-electron tomography of microtubules assembled from purified porcine brain tubulin in the presence of GTP | https://www.ebi.ac.uk/empiar/EMPIAR-11253/ | Electron Microscopy Public Image Archive, EMPIAR-11253 |
| Guyomar C, Bousquet C, Ku S, Heumann J, Guilloux G, Gaillard N, Heichette C, Duchesne L, Steinmetz MO, Gibeaux R, Chrétien D | 2022 | Cryo-electron tomography of microtubules assembled in *Xenopus* egg cytoplasmic extracts | https://www.ebi.ac.uk/empiar/EMPIAR-11263/ | Electron Microscopy Public Image Archive, EMPIAR-11263 |
| Guyomar C, Bousquet C, Ku S, Heumann J, Guilloux G, Gaillard N, Heichette C, Duchesne L, Steinmetz MO, Gibeaux R, Chrétien D | 2022 | Cryo-electron tomography of microtubules assembled from purified porcine brain tubulin in the presence of GMPCPP | https://www.ebi.ac.uk/empiar/EMPIAR-11264/ | Electron Microscopy Public Image Archive, EMPIAR-11264 |

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
