## [Editor Report]

This study presents an important finding on the assembly of microtubules in vitro, revealing structural defects accumulation in the lattice especially at the seam, where tubulin mediates lateral interactions. These defects appear at a low frequency in *Xenopus* egg cytoplasmic extracts, suggesting that cellular components control the microtubule lattice. The evidence supporting the conclusions is compelling, with rigorous cryo-electron tomography and image analysis. The work will be of broad interest to cell biologists and biochemists working on microtubules.

---

## [Decision Letter]

**Decision letter after peer review:**

Thank you for submitting your article "Changes in seam number and location induce holes within microtubules assembled from porcine brain tubulin and in *Xenopus* egg cytoplasmic extracts" for consideration by *eLife*. Your article has been reviewed by 3 peer reviewers, and the evaluation has been overseen by a Reviewing Editor and Vivek Malhotra as the Senior Editor. The following individual involved in review of your submission have agreed to reveal their identity: Khanh Huy Bui (Reviewer #1).

Essential revisions:

1) The technical concerns regarding the cryo-tomography data should be addressed fully.

2) A major concern to be addressed, was about how generalizable the findings were and whether they were the result of the quality of tubulin used, the protocol and conditions for polymerization or whether the results observed were linked to the inherent properties of tubulin. The authors should gather to increase the confidence that findings are more generalizable. it is important to know whether their findings will be more generally applicable. A number of suggestions have been listed by reviewer 3 to address this important point.

*Reviewer #1 (Recommendations for the authors):*

My major comments are:

In terms of visualization, I found the cyan and yellow balls extremely hard to correlate to the seam location. And I am saying this as someone who looks at microtubules very frequently and is extremely familiar with the A-lattice, B-lattice and the seam. It is confusing with the colour in Figure 1, where α, β-tubulin and kinesin are cyan, yellow and orange colour. The cyan ball is not at the place of the α-tubulin and in Figure 1, α-tubulin and kinesin do definitely not form a ring as shown in Figure 2B. Choosing to place the ball on only kinesin should be good and less confusing. Also, it will be a lot easy if the authors have a model of the microtubule with corresponding seams on top of the density map for aided visualization like in Figure 4C.

Figure 2B with the unrolled helix is also very confusing showing the seam in a horizontal position while in the map, the seam is in the longitudinal position.

The technical question:

– In the 14_3 microtubule in Figure 2A, what is the skew angle of the PF from the experimental data? According to the senior author's old paper (Chretien et al. 1998), the skew angle is -0.75 degrees. If it is true, the averaging of a 1390.4 nm long microtubule (~170 subtomograms) should cover well the missing wedge. I did a quick measure of the skew angle from the map EMD_15735, the skew angle measured is -0.46 (biased by the missing wedge from the map). Even with this value, the map should not contain the missing wedge. However, the map still contains the missing wedge effect. To make it transparent, it would be nice to plot the alignment of the subtomogram as the model in the microtubule to show either the accuracy of the alignment as well as the skew visualized.

– One of the weaknesses of the analysis of averaging a short segment and trying to analyze the subtomogram averaging and interpreting the seam is that the assumption of 100% kinesin decoration. If there is a partial decoration of kinesin, can the analysis stand? One of the fair ways to analyze is to perform a single kinesin/tubulin region alignment only and then plot the alignment result such as in this study from the Briggs lab (Figure 3, Faini et al. Science, 2012). Or perhaps a 3D classification to supervised/unsupervised classify regions of microtubule with a non-missing wedge reference from different microtubules to unbiasedly analyze the data vs. partitioning the microtubule into different segments.

– Also, looking at the map emd-15737 and the corresponding model, one of the seams is in close proximity to another microtubule. They seem to even touch. This should lead to bias in the analysis. It would be nice to have any seam analysis without this bias.

*Reviewer #2 (Recommendations for the authors):*

– The conclusion that they observe different protofilament, helical start and seam numbers in all conditions (L25-26, L71), is not supported by the data. All microtubules polymerised in cell extract stimulated by RanQ69L have a single seam and 13_3 protofilament architecture (shown in Figure supplement 6). These RanQ69L data are provided as a control for use of 5% DMSO for stimulation of microtubule polymerization in *Xenopus* extract. There is not enough data here to exclude the possibility that DMSO has affected the protofilament and seam number of microtubules. The RanQ69L-stimulated microtubules do contain instances of seam movement and so the main conclusions regarding the prevalence and importance of lattice-type transitions are well supported. To address this, the two datasets should be reported on separately and the differences discussed.

– The data show the variation in lattice architecture in *Xenopus* extract microtubules is very low, but this is not clear from the figures. A graph showing comparison of the different parameters measured should be included in the main figures (possibly Figure 8) to enable better visual comparison. These could include frequency of microtubule architecture (pf and start number), lattice transitions, single seam microtubules and microtubules with seam number change.

– I agree that at the sites of lattice transition there is likely a hole caused by absence of an odd number of tubulin subunits. However, the method is not able to determine their exact size or position as each segment is an average structure of 20-40 tubulin dimers. The model presented in Figure 4C does not reflect that. Could a model for only the S2 region also be shown (full length, as in the panel in Figure 4A)? The ambiguous region could be denoted by grey tubulin dimers to show the uncertainty in their exact position. The model should also be rotated to show pfa 1-4 in the center rather than offset to the left.

– The authors discuss the implications of observing mixed AB-lattices for theories about growth of microtubules via annealing of short sections or addition of subunits at the ends. These lattice discontinuities could also arise from imperfect repair of the microtubule wall after damage and this scenario should also be discussed.

– In places (e.g. L 73-74), the authors state their results show microtubules in cells are more regular and therefore their growth is tightly regulated. The consistency of protofilament number for cellular microtubules is already well known. This manuscript contains interesting new data on how the lattice can vary even when protofilament number is controlled. These statements and those in the discussion (L266-283) should be more tightly focussed on the control of A/B-type lattice formation rather than protofilament number.

*Reviewer #3 (Recommendations for the authors):*

The study by Guyomar and colleagues uses cryo-ET and subtomogram averaging to investigate the structural plasticity of microtubules assembled in vitro from purified porcine brain tubulin and from *Xenopus* egg extracts in which polymerization was initiated either by addition of DMSO or by adding a constitutively active Ran. These show that the microtubule lattice is plastic with frequent protofilament changes and containing multiple seams. A model is proposed for microtubule polymerization whereby these lattice discontinuities/defects are introduced due to the addition of tubulin dimers through lateral contacts between α and β tubulin, thus creating gaps in the lattice and shifting the seam. The structural plasticity of microtubules has been long appreciated and has returned to the fore in the last couple of years. This study provides a quantitative view of this phenomenon and clearly demonstrates the high degree of promiscuity in the polymerization of tubulin in vitro. However, I have reservations as to the generalizability and broader significance of the results and have several concerns detailed below.

1. Kinesin is used in these experiments for technical reasons, but kinesin itself modifies the lattice, so can the effects observed be exacerbated by the presence of the motor?

2. While it is true that older studies were limited methodologically from looking more quantitatively at pf changes, a very recent study from the Carter lab (Foster et al. JCB 2021) has looked at protofilament number changes in two different types of neurons and reported that such changes were extremely rare (and not at all present in DRG neurons which had exclusively 13 pf microtubules). This study is not at all mentioned or cited in this manuscript, but it seems appropriate that the authors address it in their Discussion.

3. It is maybe notable that in an intact cell pf changes are rare (as documented by old and new studies), but more frequent in the *Xenopus* extracts and even more so in the polymerized brain tubulin (~an order of magnitude higher than in the extract). Could these defects be due to denatured tubulin? This could be more pronounced in the purified tubulin sample which could have denatured/aggregated tubulin that would incorporate non-canonically in the lattice and possibly fall off and give rise to defects? Can the authors show that these numbers hold with tubulin purified from a different prep or different procedure than the batch they used? Or is the frequency of these defects a reflection of the integrity/quality of the tubulin used? Or ice thickness?

4. Are the observed defects also recapitulated if they use templated polymerization from seeds? Axonemes for example?

5. The tubulin concentration used is very high – 40 microM. Tubulin nucleates at lower concentrations and in vivo the concentration of available tubulin for polymerization is much lower (~5 microM). Do these defects show the same frequency when polymerization is carried out at lower concentrations?

6. Samples are diluted, 50X in some instances before addition to the grid – could these dilutions leads to the defects observed? Growth on the grid from a template would resolve this issue.

7. The authors speculate that EB proteins could aid in more faithful polymerization – it is though worth keeping in mind that cells do just fine without EBs, so one would think that the effects on the lattice structure would be minor or conversely the defects are not particularly relevant functionally.

All in all this is a well-executed study that shows that tubulin in vitro can be promiscuous in its polymerization properties, but it is not clear to me how generalizable the defect frequencies they find are and it is likely that they will vary with the polymerization conditions, quality of tubulin etc. It seems that a more thorough analysis would be warranted to examine how their findings hold in different assembly conditions.

---

## [Author Response]

Reviewer #1 (Recommendations for the authors):My major comments are:In terms of visualization, I found the cyan and yellow balls extremely hard to correlate to the seam location. And I am saying this as someone who looks at microtubules very frequently and is extremely familiar with the A-lattice, B-lattice and the seam. It is confusing with the colour in Figure 1, where α, β-tubulin and kinesin are cyan, yellow and orange colour. The cyan ball is not at the place of the α-tubulin and in Figure 1, α-tubulin and kinesin do definitely not form a ring as shown in Figure 2B. Choosing to place the ball on only kinesin should be good and less confusing. Also, it will be a lot easy if the authors have a model of the microtubule with corresponding seams on top of the density map for aided visualization like in Figure 4C.

As quoted in our manuscript (legend of Figure 2), cyan spheres have been placed between kinesin densities and yellow spheres on top of them. While this reminds that kinesin binds to β tubulin, explaining why we kept the same colors as in the atomic models, it is by no way a superposition of the spheres with the α- and β-tubulin subunits. Indeed, if one had to place the atomic models of tubulin in the maps, they would be slightly longitudinally displaced with respect to the positions of the spheres (this is apparent in Figure 1C), but this does not change their register within the microtubule lattice. Hence, the objective of this representation was uniquely to describe the organization of the underlying αβ-tubulin lattice, and thus end-up with simple models that we could quantify. To make it clear, we have added the following sentence in the result section (L97-101):

“To model the underlying αβ-tubulin heterodimer lattice, we placed yellow spheres onto kinesin densities and cyan spheres in between. While the cyan and yellow spheres are not strictly placed on top of the α- and β-subunits, respectively (Figure 1C), this simplified modeling allowed us to describe their underlying organization within microtubules (Figure 2B-C).”

Finally, it is unclear to us how we could have modelled the αβ-tubulin lattice organization by placing spheres only on kinesin densities.

Figure 2B with the unrolled helix is also very confusing showing the seam in a horizontal position while in the map, the seam is in the longitudinal position.

This is just a basic geometric transformation that saves space on the figures, and allowed us to present all our data (938 individual 3D reconstructions as a reminder) in graphical format in the Figure supplements for fast and simple assessment of microtubule lattice heterogeneity. E.g., comparison of Figure supplements for GDP-microtubules and *Xenopus* egg extracts shows at a first glance that these latter are much more regular.

The technical question:– In the 14_3 microtubule in Figure 2A, what is the skew angle of the PF from the experimental data? According to the senior author's old paper (Chretien et al. 1998), the skew angle is -0.75 degrees. If it is true, the averaging of a 1390.4 nm long microtubule (~170 subtomograms) should cover well the missing wedge. I did a quick measure of the skew angle from the map EMD_15735, the skew angle measured is -0.46 (biased by the missing wedge from the map). Even with this value, the map should not contain the missing wedge. However, the map still contains the missing wedge effect. To make it transparent, it would be nice to plot the alignment of the subtomogram as the model in the microtubule to show either the accuracy of the alignment as well as the skew visualized.

We suppose that the reasoning of the referee was the following: To cover the missing prism of data, sub-volumes covering the missing angular ranges must be averaged. In the present case, the missing wedge of data was 66.93° for the *a*-tilt axis (angular range -52.96°, +60.11°) and 72.80° for the *b*-tilt axis (angular range -50.33°, +56.27°). Since the microtubule was roughly parallel to the *a*-tilt axis (see Figure A on the right), we will consider its missing wedge.

The length required to cover the missing wedge can be expressed as L=N*dx*w/abs(tan(q)*360), where w is the missing wedge (in degrees), dx is the separation between protofilaments (5.15 nm (Chrétien and Wade, 1991)), N the number of protofilaments, and q the protofilament skew angle. The protofilament skew angle measured directly on the raw data was -0.64° (-0.75° is a theoretical prediction), which gives a length L = 1200 nm to fill in the missing wedge with new data. Hence, as expected by the referee, the length of the microtubule (1390 nm) was sufficient (although according to our calculations a protofilament skew angle of -0.46° would have required a 1670 nm long microtubule. Hence, the referee might have used calculations different from ours).

In Author response image 1, we compare a 13_3 (MT4 in Figure 2—Figure supplement 3) with the 14_3 microtubule of Figure 2 (MT3 in Figure 2—Figure supplement 3). The two microtubules are oriented close to the parallel of the *a*-tilt axis (Author response image 1 A). In Author response image 1 B, we compare projections over 10 slices of the sub-tomogram averages of the 13_3 (left) and 14_3 (right) microtubules, together with their corresponding Fourier transforms in Author response image 1 C. Since the protofilaments do not rotate in the 13_3 microtubules, the elongation of densities in the direction of the missing wedge is clearly apparent and the missing wedge is not compensated in its FFT. In the case of the 14_3 microtubules, the protofilaments are better resolved on the edges, and its Fourier transform shows additional data into the missing wedge, although with less densities than in the original regions.

**Author response image 1. sa2fig1:** 

To our opinion, this is likely a consequence of an uneven sampling of Fourier space after adding a limited number of views into the missing wedge compared to the original regions that have been reinforced by the averaging procedure. To visualize this effect, we have made a simple simulation with a Fourier transform of a tomogram (Author response image 1 D), applied clockwise rotations every 4° over 60° (Author response image 1 E) and 180° (Author response image 1 F). This simulation reproduces fairly our sub-tomographic approach, since we add progressively views into the missing wedge by following individual protofilaments along the microtubule. On the right are profile densities along the yellow circle. It can be seen that while data are incorporated into the missing wedge in (E), their intensities are lower than in the original regions of the FFT. Indeed, at least a 180° rotation of the protofilaments around the microtubule axis is necessary to evenly fill the Fourier space (F), a prerequisite to obtain a fully symmetric map devoid of missing wedge artefacts. In theory, PEET is supposed to compensate for unequal distribution of densities in Fourier space, but this may only be partial as observed in our data. Nevertheless, these missing wedge artefacts present in the full volumes, and also in the segments, do not modify the positions of the kinesin-motor domain densities, and have no impact on the characterization of microtubule lattice organization in the microtubules.

– One of the weaknesses of the analysis of averaging a short segment and trying to analyze the subtomogram averaging and interpreting the seam is that the assumption of 100% kinesin decoration. If there is a partial decoration of kinesin, can the analysis stand?

There is no assumption on the level of kinesin decoration in our analysis. The result of a partial decoration will be a decrease in kinesin motor-domain density in the sub-tomogram averages, but by no means a change in their registry along individual protofilaments.

One of the fair ways to analyze is to perform a single kinesin/tubulin region alignment only and then plot the alignment result such as in this study from the Briggs lab (Figure 3, Faini et al. Science, 2012). Or perhaps a 3D classification to supervised/unsupervised classify regions of microtubule with a non-missing wedge reference from different microtubules to unbiasedly analyze the data vs. partitioning the microtubule into different segments.

We sincerely doubt than an approach based on a single kinesin/tubulin alignment would be successful, in particular due to the imaging artefacts and the interaction of microtubules with the air-water interface (see Figure 6). Nevertheless, we acknowledge that more sophisticate methods than ours can be imagined to analyze the changes in lattice types within individual microtubules. One of the main limitations of our approach is that, by averaging over short segments, we likely underestimate the lattice-type transition frequencies, and hence the number of holes within microtubules (this is discussed in the manuscript). New analytical methods may tackle this issue. Developing these is however beyond the scope of this paper and our method has the advantage to be both sufficiently efficient and robust to reveal the so far unexpected lattice heterogeneity of individual microtubules and the differences between the in vitro and the *ex cellulo* contexts. Since the tomograms have now been deposited onto the EMPIAR (new Supplementary file 2), anybody is free to try new processing strategies, and we would be of course delighted to learn that more powerful methods than ours have been developed.

– Also, looking at the map emd-15737 and the corresponding model, one of the seams is in close proximity to another microtubule. They seem to even touch. This should lead to bias in the analysis. It would be nice to have any seam analysis without this bias.

During sub-tomogram averaging, a cylindrical mask is created around the microtubule of interest (and one also inside, see Author response image 2 on the right: Reference), so that densities arising from microtubules in close proximity have little or no influence on the registering procedure (see the online tutorial referenced in the Materials and methods section under Sub-tomogram averaging). To make this clear, we have added the following sentence (L495-496):

'Inner and outer cylindrical masks were used to isolate the microtubule wall densities'.

The additional densities coming from the neighborhood microtubule visible on the right side of the Final map arise only at the end of the process during back-projection of the sub-volumes, and hence do not produce any bias in the computation of the final sub-tomogram averages. A detailed protocol to perform SSTA has been written and will be submitted following publication of the present manuscript. Since our tomograms and models have been deposited onto the EMPIAR database, all sub-tomogram averages used in this study can be redone, including numerous examples of microtubules with seams not touching other microtubules.

Overall, we have the feeling that the referee was highly concerned with the missing wedge artefacts (a missing prism in most of our data, since we essentially used dual-axis cryo-ET). The main effect of missing data at high angle (whether it is a missing wedge or a prism), is an elongation of structures along the beam direction (Guesdon et al., 2013; Mastronarde, 1997). This is clearly a limitation when one tries to analyze microtubule edges that have been severely smoothed by this imaging artefact, explaining why some averaging over short segments is still necessary. However, this artefact does not modify the positions of the kinesin motor domains in the final volumes, allowing us to describe their organization in the segmented sub-tomogram averages.

Reviewer #2 (Recommendations for the authors):Main points:– The conclusion that they observe different protofilament, helical start and seam numbers in all conditions (L25-26, L71), is not supported by the data. All microtubules polymerised in cell extract stimulated by RanQ69L have a single seam and 13_3 protofilament architecture (shown in Figure supplement 6). These RanQ69L data are provided as a control for use of 5% DMSO for stimulation of microtubule polymerization in *Xenopus* extract. There is not enough data here to exclude the possibility that DMSO has affected the protofilament and seam number of microtubules. The RanQ69L-stimulated microtubules do contain instances of seam movement and so the main conclusions regarding the prevalence and importance of lattice-type transitions are well supported. To address this, the two datasets should be reported on separately and the differences discussed.

In these sentences (L25-26, L71), we indeed made reference to *Xenopus* egg cytoplasmic extracts as a whole, independently of whether microtubule aster formation was stimulated by DMSO or Ran. Hence, the referee is right and we changed the phrase by (L25-26): 'We find that in almost all conditions the seam number and location vary within individual microtubules, leaving holes of one to a few subunits in size within their wall.'. This is also corrected in L71. In the Ran experiments, we simply asked whether lattice-type transitions were present like in DMSO egg extracts, which was indeed the case. Following the referee's suggestions, we have modified the end of the result section as follows L221-224:

“In addition, variations in protofilament and helix-start numbers were also observed such as 12_2, 12_3, 13_4 and 14_3 microtubule-lattice regions, but uniquely in the Xenopus-DMSO sample (Figure 9A-D, Figure 9—Video 1, Figure 10A, Table 2)', and (L229-233): ' Microtubules with protofilament numbers different than 13 were not observed in the *Xenopus*-Ran sample (Figure 8—figure supplement 5, Figure 10B, Table 2). Hence, we cannot exclude the possibility that DMSO induced the formation of these microtubules in Xenopus egg cytoplasmic extracts, and it remains to be determined whether they also occur in intact Xenopus eggs.”

– The data show the variation in lattice architecture in *Xenopus* extract microtubules is very low, but this is not clear from the figures. A graph showing comparison of the different parameters measured should be included in the main figures (possibly Figure 8) to enable better visual comparison. These could include frequency of microtubule architecture (pf and start number), lattice transitions, single seam microtubules and microtubules with seam number change.

We have created a new Figure 10 that includes for each condition the lattice types (Figure 10A), the protofilament and helix-start numbers (Figure 10B), the seam number (Figure 10C) and the lattice type transitions (Figure 10D). This figure is now referenced in L254-255:

“Third, the lattice-type transition frequency remains low with respect to the number of tubulin heterodimers within microtubules (Figure 10C-D—Table 1).”

– I agree that at the sites of lattice transition there is likely a hole caused by absence of an odd number of tubulin subunits. However, the method is not able to determine their exact size or position as each segment is an average structure of 20-40 tubulin dimers. The model presented in Figure 4C does not reflect that. Could a model for only the S2 region also be shown (full length, as in the panel in Figure 4A)? The ambiguous region could be denoted by grey tubulin dimers to show the uncertainty in their exact position. The model should also be rotated to show pfa 1-4 in the center rather than offset to the left.

The figure legend of Figure 4C specifically mentions that the transition regions require an offset of *'at least one monomer'* such as presented in the models, which implies that larger holes can be present. To make it even clearer, we have modified the figure legend as follow (L798-801):

“The transition from 5 seams in S1 to 3 seams in S3 requires an offset of at least one monomer (red stars) in the protofilaments 3 and 4 of S2, although larger holes of an odd number of subunits could be present.”

The orientation of the models was chosen so that the change from 3 to 5 seams could be visualized (dotted lines). Orienting the models as suggested by the referee would unfortunately hide the two seams on the right.

– The authors discuss the implications of observing mixed AB-lattices for theories about growth of microtubules via annealing of short sections or addition of subunits at the ends. These lattice discontinuities could also arise from imperfect repair of the microtubule wall after damage and this scenario should also be discussed.

It is difficult to imagine how a local repair could induce a long-range shift of protofilament register within microtubules, such as observed in the vast majority of the microtubules in the present study. Hence, we feel uncomfortable to discuss this issue in the current version of our manuscript. Such issue might be addressed later when more data will be available on different systems and assembly conditions. For example, it would be interesting to perform repair experiments with gold-labelled tubulin to observe its incorporation at repair sites.

– In places (e.g. L 73-74), the authors state their results show microtubules in cells are more regular and therefore their growth is tightly regulated. The consistency of protofilament number for cellular microtubules is already well known. This manuscript contains interesting new data on how the lattice can vary even when protofilament number is controlled. These statements and those in the discussion (L266-283) should be more tightly focussed on the control of A/B-type lattice formation rather than protofilament number.

By contrast the referee's claim stating that the protofilament number for cellular microtubules is already well known, there is an abundant literature showing exceptions to the '13 protofilament rule' in different cell types and species (see Chrétien and Wade, 1992; Chaaban and Brouhard, 2017, for instance). While most previous data on cell microtubule architecture have been gathered from thin sections of specimens embedded in resin and further stained with the tannic-acid method, the possibility to characterize long stretches of cellular microtubules offered by the method of cryo-electron tomography such as performed by (Foster et al., 2021), will certainly provide new and interesting information concerning this issue.

But most importantly, and as we discuss it in our manuscript, we know very little concerning the organization of the ab-tubulin heterodimers within cellular microtubules. The vast majority of the microtubules in *Xenopus* egg cytoplasmic extracts are 13_3 with a B-type lattice and a unique seam, while in vitro, this microtubule configuration can incorporate multi-seams. Hence, we believe that it is fair to discuss these two features (lattice type and protofilament number) in parallel. We specifically focus on EBs because there are involved in those two aspects: they bind in between protofilaments organized according to a B-type lattice and they also favor the formation of 13_3 microtubules. Therefore, we wish to maintain this part of the discussion in its current form.

Reviewer #3 (Recommendations for the authors):The study by Guyomar and colleagues uses cryo-ET and subtomogram averaging to investigate the structural plasticity of microtubules assembled in vitro from purified porcine brain tubulin and from *Xenopus* egg extracts in which polymerization was initiated either by addition of DMSO or by adding a constitutively active Ran. These show that the microtubule lattice is plastic with frequent protofilament changes and containing multiple seams. A model is proposed for microtubule polymerization whereby these lattice discontinuities/defects are introduced due to the addition of tubulin dimers through lateral contacts between α and β tubulin, thus creating gaps in the lattice and shifting the seam. The structural plasticity of microtubules has been long appreciated and has returned to the fore in the last couple of years. This study provides a quantitative view of this phenomenon and clearly demonstrates the high degree of promiscuity in the polymerization of tubulin in vitro. However, I have reservations as to the generalizability and broader significance of the results and have several concerns detailed below.1. Kinesin is used in these experiments for technical reasons, but kinesin itself modifies the lattice, so can the effects observed be exacerbated by the presence of the motor?

See point #5 in our answer to the referee's Public Review. Yet, it remains unclear to us which "modification" the referee refers to. Peet et al. (2018) used fluorescence microscopy to show that the motor domain of kinesin 1 can expand the lattice by ~1.6% (Peet et al., 2018), but these data do not seem to be substantiated by the cryo-EM experiments listed in Figure 3 of (Manka and Moores, 2018). Nevertheless, even if kinesin motor domains expand the microtubule lattice, we do not imagine how this could be related to the changes in registry of the ab-tubulin heterodimers that we observe within preformed microtubules. Other modifications reported recently concern the removal of tubulin dimers by full-length kinesin during his walk onto microtubules (Budaitis et al., 2021; Kuo et al., 2022; Sabo and Lansky, 2022; Triclin et al., 2021). While this process must create holes within microtubules, here, we used only the motor domain of kinesin 1, which is non processive. Of course, we could answer more precisely if the referee could provide adequate references to his claim that 'kinesin itself modifies the lattice'.

2. While it is true that older studies were limited methodologically from looking more quantitatively at pf changes, a very recent study from the Carter lab (Foster et al. JCB 2021) has looked at protofilament number changes in two different types of neurons and reported that such changes were extremely rare (and not at all present in DRG neurons which had exclusively 13 pf microtubules). This study is not at all mentioned or cited in this manuscript, but it seems appropriate that the authors address it in their Discussion.

See point #2 in our answer to the referee's Public Review. Nevertheless, the referee is right concerning protofilament number transitions along individual microtubules, and we now cite (Foster et al., 2021) in the Ideas and speculation section (L352-354):

“These numbers can vary within individual microtubules (Chrétien et al., 1992; Foster et al., 2021), necessarily leaving holes inside their lattice (Schaedel et al., 2019; Théry and Blanchoin, 2021)."

3. It is maybe notable that in an intact cell pf changes are rare (as documented by old and new studies), but more frequent in the *Xenopus* extracts and even more so in the polymerized brain tubulin (~an order of magnitude higher than in the extract).

Apart from the study of Foster et al. (2021) cited above, we do not know other studies where changes in protofilament numbers along individual microtubules have been documented in intact cells. In interphasic *Xenopus* egg extracts, it was reported to be 0.04 µm^-1^ (calculated from data in Table II of (Chrétien et al., 1992)), which is the same order of magnitude to what we observe here in metaphase arrested egg extracts (~0.1 µm^-1^). However, we cannot tell whether these transitions are more frequent than in intact cells, since we could not find such data in the work of Foster et al. (2021) or in any other previous studies. We would of course be interested to know the old and new references suggested by the referee so that we can discuss them in our manuscript. Alternatively, if by 'protofilament changes' the referee refers to the range of protofilament numbers different than 13 in cells, there is an abundant literature on this subject reviewed recently by (Chaaban and Brouhard, 2017) (see also our answer to reviewer #2 concerning this issue).

Could these defects be due to denatured tubulin? This could be more pronounced in the purified tubulin sample which could have denatured/aggregated tubulin that would incorporate non-canonically in the lattice and possibly fall off and give rise to defects?

At the molecular level, changes in lattice types involve switches between homotypic (αα, ββ) and heterotypic (αβ, βα) lateral interactions in adjacent protofilaments. None of these are abnormal interactions (i.e., 'non-canonical'), since they occur naturally in microtubules. By analogy with defects found in crystals and nanotubes for instance, the term 'defect' is a geometrical characteristic that must be understood as a local modification in the registry of the ab-heterodimers within the microtubule lattice, and by no way a 'defect' in the tubulin molecule by itself. Hence, there is no need to speculate that denatured tubulin is at the origin of such transitions that we also find in a cytoplasmic context.

Nevertheless, to address the referee's concern, we must quote that we use very classical protocols to purify tubulin and assemble microtubules, similar to those used by many laboratories in the field. The present batch of tubulin was prepared according to (Castoldi and Popov, 2003) followed by a cycling step to remove free GTP (Ashford and Hyman, 2006) and was stored at -80 °C until use. Upon tawing, tubulin is diluted at the right concentration, incubated with the nucleotide of interest (here, GTP at 1 mM or GMPCPP at 0.1 mM), and subjected to a high-speed run in an airfuge to remove denaturated tubulin that typically forms aggregates upon tawing. Tubulin is a very labile protein that denaturates with time at 4 °C (Ashford and Hyman, 2006), implying that experiments should be performed quickly after tawing. In addition, tubulin denaturation occurs inevitably during microtubule assembly (see our detailed study in (Weis et al., 2010) concerning this issue and the role of the chaperone HSP90 in protecting tubulin from thermal denaturation). However, once denaturated the protein forms aggregates, and to our knowledge, incorporation of denatured tubulin within microtubules has not been reported.

We also stress that we are one of the rare laboratories that check the level of aggregation after microtubule self-assembly, by cooling down the sample at 4 °C and measure the difference in OD between the start of the reaction and the basal level at 4 °C, see Figure 2—figure supplement 1A. In this figure, it can be seen that the level of aggregation is very low compared to the OD contributed by the microtubules at the polymerization the plateau. Nevertheless, if we follow the referee reasoning, lattice defects should accumulate at the plateau during self-assembly. Conversely, if the frequency of lattice defects scales with the microtubule growth rate, it should be very high during the sigmoid phase of microtubule self-assembly, and then decrease progressively at the plateau due to microtubule dynamic instability. Experiments are planned to test this latter hypothesis (see also below), which will indirectly address the referee's concerns about tubulin denaturation.

Can the authors show that these numbers hold with tubulin purified from a different prep or different procedure than the batch they used? Or is the frequency of these defects a reflection of the integrity/quality of the tubulin used? Or ice thickness?

See point #3 in our answer to the referee's Public Review. Please also see a Figure showing the comparison between the seam distribution obtained by Debs et al. (2020) on Taxol stabilized self-assembled microtubules, and the one we obtained with self-assembled GDP-microtubules (Figure 10C, dark-blue bars). The fact that multi-seams have been observed by several group since 1994, and that used different sources of tubulin and different protocols for its purification, strongly suggests that it is a general property of microtubule assembly from purified tubulin. Hence, our results are not only in perfect line with these previous observations, but go beyond since we find that the number and location of seams vary within individual microtubules. This was not described before because none of these previous studies addressed the structural heterogeneity of individual microtubule lattices. In addition, it must be stressed that in all these previous studies, Taxol was used to assemble and/or stabilize the microtubules, raising the concern that the drug could have induced multi-seams. Here, we used classical self-assembly conditions in the presence of GTP to show that the formation of multi-seams is inherent to the tubulin polymerization reaction.

Concerning ice thickness, we cannot imagine how this could modify the registry of the αβ-tubulin heterodimer within the microtubule lattice.

4. Are the observed defects also recapitulated if they use templated polymerization from seeds? Axonemes for example?

This is a study that we are planning to perform to investigate the effect of tubulin concentration, and hence growth rate, on lattice type transition frequency, similar to the one described in (Schaedel et al., 2019) where we investigated changes in protofilament numbers as a function of tubulin concentration. These experiments will take time and go clearly beyond the scope of the present study.

5. The tubulin concentration used is very high – 40 microM.

See point #1 in our answer to the referee's Public Review. Maybe the referee was misled by our sentence in L139-141 were we quoted 'Direct visualization of holes within microtubules self-assembled at high tubulin concentration (40 µM) in the presence of GTP was hampered by the high background generated by free tubulin in solution.' We used the term "high" by comparison to the GMPCPP conditions where tubulin can assemble at a lower concentration (10 µM). To correct this, we now quote (L139-141):

“Direct visualization of holes within microtubules self-assembled at a tubulin concentration of 40 µM in the presence of GTP was hampered by the high background generated by free tubulin in solution.”

Tubulin nucleates at lower concentrations and in vivo the concentration of available tubulin for polymerization is much lower (~5 microM).

Concerning the CC to overcome the nucleation barrier with mammalian brain tubulin, see our answer above. To our knowledge, the accepted value for tubulin concentration in cells from vertebrates is closer to ~20 µM e.g., it was estimated to ~22 µM in 3T3 cells (Hiller and Weber, 1978) and ~24 µM in egg extracts (Gard and Kirschner, 1987). We hypothesize that the value of 5 µM quoted by the referee is derived from measurements in *S. pombe* (Loiodice et al., 2019) that contains a very limited number of microtubules (Höög et al., 2007), and may not be taken as representative of eucaryotic cells in vertebrates from instance. Nevertheless, variations in tubulin concentration likely occur between cell types in a same species, and also between species. Yet, microtubule nucleation and assembly in cells is regulated by a vast array of macromolecular assemblages and proteins such as the g-TuRC and MAPs. Here, we used a tubulin concentration ~twice that reported in cells, which remains close to physiological conditions, at least the same order of magnitude. Therefore, it cannot be quoted as 'very high'. Noticeably, it may turn out to be crucial for cells to have a tubulin concentration close to its CC so that self-assembly does not occur, and thus ensure that microtubule polymerization starts from nucleating centers such as the centrosome.

Do these defects show the same frequency when polymerization is carried out at lower concentrations?

This is a good point that we plan to examine in details. The general question behind is whether the lattice-type transition frequency increases with the microtubule growth rate. We already showed that this is the case for the protofilament number transition frequency, and that as the growth rates increases, a wider range of protofilament numbers is formed (Schaedel et al., 2019). We thus speculate that this will also hold true for the lattice-type transition frequency, but this remains to be investigated.

6. Samples are diluted, 50X in some instances before addition to the grid – could these dilutions leads to the defects observed? Growth on the grid from a template would resolve this issue.

See point #4 in our answer to the referee's Public Review. In addition, it is unclear to us how seeded assembly would avoid this phenomenon. Microtubules have been reported to transition in protofilament number, even when nucleated by centrosomes (Schaedel et al., 2019). Hence, we anticipate that transitions in lattice-types will follow the same trends once nucleated from seeds. But as stated above, this is a study that we plan to perform.

7. The authors speculate that EB proteins could aid in more faithful polymerization – it is though worth keeping in mind that cells do just fine without EBs, so one would think that the effects on the lattice structure would be minor or conversely the defects are not particularly relevant functionally.

By contrast to the referee's claims, there is an abundant literature that demonstrated that perturbing EBs has strong consequences on cell functions. To cite a few, in (Draviam et al., 2006), RNAi of EB1 resulted in chromosome missegregation at anaphase. In (Komarova et al., 2009), shRNA of EB1 and EB3 promoted persistent microtubule growth, whereas in the presence of purified tubulin they increased the catastrophe frequency. Noticeably, the authors report that a triple KO of EB1, EB2 and EB3 was incompatible with cell viability. In (Thomas et al., 2016), siRNA of EB1 resulted in chromosome misalignment due to a decrease in Ska1 (spindle-kinetochore-associated protein) association with microtubules. In (Yang et al., 2017), disruption of EB1, EB2 and EB3 had little impact on cell division, but perturbed the organization of non centrosomal microtubules, leading to a compaction of the Golgi as well as defects in cell migration. In our hands, depleting EBs from *Xenopus* egg extracts results in a strong decrease in the size of microtubule asters, and a deregulation of microtubule protofilament number (ongoing experiments). This is perfectly in line with previous results by us and others showing that EBs favor 13_3 microtubules (des Georges et al., 2008; Vitre et al., 2008).

All in all this is a well-executed study that shows that tubulin in vitro can be promiscuous in its polymerization properties, but it is not clear to me how generalizable the defect frequencies they find are and it is likely that they will vary with the polymerization conditions, quality of tubulin etc. It seems that a more thorough analysis would be warranted to examine how their findings hold in different assembly conditions.

Generalization of our work seems to be a major concern of the referee. Conversely, we argue, based on a robust bibliography background, that multi-seams have been observed by many others in the past, and hence with different tubulin batches and widely different assembly conditions (see point #3 in our answer to the referee's Public Review). Finally, we acknowledge the fact that our discovery opens new perspectives such as studies on the effect of microtubule growth rate on lattice type transition frequency and the effect of MAPs (including EBs) on their structural integrity. These are all ongoing experiments, but that go clearly outside the scope of the current study and that will require more lengthy and detailed studies.

References

Ashford, A. J., and Hyman, A. A. (2006). Chapter 22—Preparation of Tubulin from Porcine Brain. In J. E. Celis (Éd.), *Cell Biology (Third Edition)* (p. 155‑160). Academic Press. https://doi.org/10.1016/B978-012164730-8/50094-0

Budaitis, B. G., Badieyan, S., Yue, Y., Blasius, T. L., Reinemann, D. N., Lang, M. J., Cianfrocco, M. A., and Verhey, K. J. (2021). A kinesin-1 variant reveals motor-induced microtubule damage in cells [Preprint]. Cell Biology. https://doi.org/10.1101/2021.10.19.464974

Castoldi, M., and Popov, A. V. (2003). Purification of brain tubulin through two cycles of polymerization–depolymerization in a high-molarity buffer. *Protein Expression and Purification*, *32*(1), 83‑88. https://doi.org/10.1016/S1046-5928(03)00218-3

Chaaban, S., and Brouhard, G. J. (2017). A microtubule bestiary : Structural diversity in tubulin polymers. *Molecular Biology of the Cell*, *28*(22), 2924‑2931. https://doi.org/10.1091/mbc.E16-05-0271

Chrétien, D., Metoz, F., Verde, F., Karsenti, E., and Wade, R. (1992). Lattice defects in microtubules : Protofilament numbers vary within individual microtubules. *Journal of Cell Biology*, *117*(5), 1031‑1040. https://doi.org/10.1083/jcb.117.5.1031

Chrétien, D., and Wade, R. H. (1991). New data on the microtubule surface lattice. *Biology of the Cell*, *71*(1‑2), 161‑174. https://doi.org/10.1016/0248-4900(91)90062-r

Debs, G. E., Cha, M., Liu, X., Huehn, A. R., and Sindelar, C. V. (2020). Dynamic and asymmetric fluctuations in the microtubule wall captured by high-resolution cryoelectron microscopy. Proceedings of the National Academy of Sciences of the United States of America, 117(29), 16976‑16984. https://doi.org/10.1073/pnas.2001546117

des Georges, A., Katsuki, M., Drummond, D. R., Osei, M., Cross, R. A., and Amos, L. A. (2008). Mal3, the Schizosaccharomyce *S. pombe* homolog of EB1, changes the microtubule lattice. *Nature Structural and Molecular Biology*, *15*(10), 1102‑1108. https://doi.org/10.1038/nsmb.1482

Dias, D. P., and Milligan, R. A. (1999). Motor protein decoration of microtubules grown in high salt conditions reveals the presence of mixed lattices. *Journal of Molecular Biology*, *287*(2), 287‑292. https://doi.org/10.1006/jmbi.1999.2597

Draviam, V. M., Shapiro, I., Aldridge, B., and Sorger, P. K. (2006). Misorientation and reduced stretching of aligned sister kinetochores promote chromosome missegregation in EB1- or APC-depleted cells. *The EMBO Journal*, *25*(12), 2814‑2827. https://doi.org/10.1038/sj.emboj.7601168

Foster, H. E., Ventura Santos, C., and Carter, A. P. (2021). A cryo-ET survey of microtubules and intracellular compartments in mammalian axons. *Journal of Cell Biology*, *221*(2), e202103154. https://doi.org/10.1083/jcb.202103154

Gard, D. L., and Kirschner, M. W. (1987). A microtubule-associated protein from *Xenopus* eggs that specifically promotes assembly at the plus-end. *The Journal of Cell Biology*, *105*(5), 2203‑2215. https://doi.org/10.1083/jcb.105.5.2203

Guesdon, A., Blestel, S., Kervrann, C., and Chrétien, D. (2013). Single versus dual-axis cryo-electron tomography of microtubules assembled in vitro : Limits and perspectives. *Journal of Structural Biology*, *181*(2), 169‑178. https://doi.org/10.1016/j.jsb.2012.11.004

Hiller, G., and Weber, K. (1978). Radioimmunoassay for tubulin : A quantitative comparison of the tubulin content of different established tissue culture cells and tissues. *Cell*, *14*(4), 795‑804. https://doi.org/10.1016/0092-8674(78)90335-5

Höög, J. L., Schwartz, C., Noon, A. T., O’Toole, E. T., Mastronarde, D. N., McIntosh, J. R., and Antony, C. (2007). Organization of Interphase Microtubules in Fission Yeast Analyzed by Electron Tomography. *Developmental Cell*, *12*(3), 349‑361. https://doi.org/10.1016/j.devcel.2007.01.020

Kikkawa, M., Ishikawa, T., Nakata, T., Wakabayashi, T., and Hirokawa, N. (1994). Direct visualization of the microtubule lattice seam both in vitro and in vivo. *The Journal of Cell Biology*, *127*(6 Pt 2), 1965‑1971. https://doi.org/10.1083/jcb.127.6.1965

Komarova, Y., De Groot, C. O., Grigoriev, I., Gouveia, S. M., Munteanu, E. L., Schober, J. M., Honnappa, S., Buey, R. M., Hoogenraad, C. C., Dogterom, M., Borisy, G. G., Steinmetz, M. O., and Akhmanova, A. (2009). Mammalian end binding proteins control persistent microtubule growth. *Journal of Cell Biology*, *184*(5), 691‑706. https://doi.org/10.1083/jcb.200807179

Kuo, Y.-W., Mahamdeh, M., Tuna, Y., and Howard, J. (2022). The force required to remove tubulin from the microtubule lattice by pulling on its α-tubulin C-terminal tail. *Nature Communications*, *13*(1), Art. 1. https://doi.org/10.1038/s41467-022-31069-x

Loiodice, I., Janson, M. E., Tavormina, P., Schaub, S., Bhatt, D., Cochran, R., Czupryna, J., Fu, C., and Tran, P. T. (2019). Quantifying Tubulin Concentration and Microtubule Number Throughout the Fission Yeast Cell Cycle. *Biomolecules*, *9*(3), 86. https://doi.org/10.3390/biom9030086

Manka, S. W., and Moores, C. A. (2018). Microtubule structure by cryo-EM : Snapshots of dynamic instability. *Essays in Biochemistry*, *62*(6), 737‑751. https://doi.org/10.1042/EBC20180031

Mastronarde, D. N. (1997). Dual-Axis Tomography : An Approach with Alignment Methods That Preserve Resolution. *Journal of Structural Biology*, *120*(3), 343‑352. https://doi.org/10.1006/jsbi.1997.3919

Mitchison, T., and Kirschner, M. (1984a). Dynamic instability of microtubule growth. *Nature*, *312*(5991), Art. 5991. https://doi.org/10.1038/312237a0

Mitchison, T., and Kirschner, M. (1984b). Microtubule assembly nucleated by isolated centrosomes. *Nature*, *312*(5991), 232‑237. https://doi.org/10.1038/312232a0

Peet, D. R., Burroughs, N. J., and Cross, R. A. (2018). Kinesin expands and stabilizes the GDP-microtubule lattice. *Nature Nanotechnology*, *13*(5), 386‑391. https://doi.org/10.1038/s41565-018-0084-4

Sabo, J., and Lansky, Z. (2022). Molecular motors : Turning kinesin-1 into a microtubule destroyer. *Current Biology*, *32*(11), R518‑R520. https://doi.org/10.1016/j.cub.2022.04.079

Schaedel, L., Triclin, S., Chrétien, D., Abrieu, A., Aumeier, C., Gaillard, J., Blanchoin, L., Théry, M., and John, K. (2019). Lattice defects induce microtubule self-renewal. *Nature Physics*, *15*(8), 830‑838. https://doi.org/10.1038/s41567-019-0542-4

Sosa, H., Hoenger, A., and Milligan, R. A. (1997). Three different approaches for calculating the three-dimensional structure of microtubules decorated with kinesin motor domains. *Journal of Structural Biology*, *118*(2), 149‑158. https://doi.org/10.1006/jsbi.1997.3851

Théry, M., and Blanchoin, L. (2021). Microtubule self-repair. *Current Opinion in Cell Biology*, *68*, 144‑154. https://doi.org/10.1016/j.ceb.2020.10.012

Thomas, G. E., Bandopadhyay, K., Sutradhar, S., Renjith, M. R., Singh, P., Gireesh, K. K., Simon, S., Badarudeen, B., Gupta, H., Banerjee, M., Paul, R., Mitra, J., and Manna, T. K. (2016). EB1 regulates attachment of Ska1 with microtubules by forming extended structures on the microtubule lattice. *Nature Communications*, *7*, 11665. https://doi.org/10.1038/ncomms11665

Triclin, S., Inoue, D., Gaillard, J., Htet, Z. M., DeSantis, M. E., Portran, D., Derivery, E., Aumeier, C., Schaedel, L., John, K., Leterrier, C., Reck-Peterson, S. L., Blanchoin, L., and Théry, M. (2021). Self-repair protects microtubules from destruction by molecular motors. *Nature Materials*, *20*(6), 883‑891. https://doi.org/10.1038/s41563-020-00905-0

Vitre, B., Coquelle, F. M., Heichette, C., Garnier, C., Chrétien, D., and Arnal, I. (2008). EB1 regulates microtubule dynamics and tubulin sheet closure in vitro. *Nature Cell Biology*, *10*(4), 415‑421. https://doi.org/10.1038/ncb1703

Weis, F., Moullintraffort, L., Heichette, C., Chrétien, D., and Garnier, C. (2010). The 90-kDa heat shock protein Hsp90 protects tubulin against thermal denaturation. *The Journal of Biological Chemistry*, *285*(13), 9525‑9534. https://doi.org/10.1074/jbc.M109.096586

Yang, C., Wu, J., de Heus, C., Grigoriev, I., Liv, N., Yao, Y., Smal, I., Meijering, E., Klumperman, J., Qi, R. Z., and Akhmanova, A. (2017). EB1 and EB3 regulate microtubule minus end organization and Golgi morphology. *The Journal of Cell Biology*, *216*(10), 3179‑3198. https://doi.org/10.1083/jcb.201701024

Zhang, R., Alushin, G. M., Brown, A., and Nogales, E. (2015). Mechanistic Origin of Microtubule Dynamic Instability and Its Modulation by EB Proteins. *Cell*, *162*(4), 849‑859. https://doi.org/10.1016/j.cell.2015.07.012

Zhang, R., LaFrance, B., and Nogales, E. (2018). Separating the effects of nucleotide and EB binding on microtubule structure. *Proceedings of the National Academy of Sciences*, *115*(27), E6191‑E6200. https://doi.org/10.1073/pnas.1802637115